# Scaling Laws for Imitation Learning in Single-Agent Games

**Jens Tuyls[1], Dhruv Madeka[2], Kari Torkkola[2], Dean P. Foster[2,4], Karthik Narasimhan[1], Sham Kakade[2,3]**
*[1]Princeton University, [2]Amazon, [3]Harvard University, [4]University of Pennsylvania*
*jtuyls@cs.princeton.edu*

**Reviewed on OpenReview:** *https://openreview.net/forum?id=qGVchjFj3a*

## Abstract

Imitation Learning (IL) is one of the most widely used methods in machine learning. Yet, many works find it is often unable to fully recover the underlying expert behavior (Wen et al., 2020; Jacob et al., 2022), even in constrained environments like single-agent games (De Haan et al., 2019; Hambro et al., 2022b). However, none of these works deeply investigate the role of scaling up the model and data size. Inspired by recent work in Natural Language Processing (NLP) (Kaplan et al., 2020; Hoffmann et al., 2022) where "scaling up" has resulted in increasingly more capable LLMs, we investigate whether carefully scaling up model and data size can bring similar improvements in the imitation learning setting for single-agent games. We first demonstrate our findings on a variety of Atari games, and thereafter focus on the extremely challenging game of NetHack. In all games, we find that IL *loss* and *mean return* scale smoothly with the compute budget (FLOPs) and are strongly correlated, resulting in power laws (and variations of them) for training compute-optimal IL agents. Finally, we forecast and train several NetHack agents with IL and find our best agent outperforms the prior state-of-the-art by 1.7x in the offline setting. Our work both demonstrates the scaling behavior of imitation learning in a variety of single-agent games, as well as helps narrow the gap between the learner and the expert in NetHack, a game that remains elusively hard for current AI systems.[1]

## 1 Introduction

While conceptually simple, imitation learning has powered some of the most impressive feats of AI in recent years. AlphaGo (Silver et al., 2016) used imitation on human Go games to bootstrap its Reinforcement Learning (RL) policy. Cicero, an agent that can play the challenging game of Diplomacy, used an IL-based policy as an anchor to guide planning (Jacob et al., 2022). Go-Explore, a method for hard-exploration problems which solved all previously unsolved Atari games, used self-imitation learning in its robustification phase (Ecoffet et al., 2021).

Despite its prevalence, several works have pointed out some of the limitations of IL. De Haan et al. (2019) and Wen et al. (2020) call out the issue of *causal confusion*, where the IL policy relies on spurious correlations to achieve high training and held-out accuracy, but performs far worse than the data-generating policy, even in single-agent Atari games. Jacob et al. (2022) have mentioned similar issues for policies learning from human games: *they consistently underperform the data-generating policy*. However, in many of these works, the role of model and data size is not deeply investigated. This is especially striking considering the increasingly impressive capabilities that recent language models have exhibited, mostly as a consequence of scale. In a series of papers trying to characterize these improvements with scale starting with Hestness et al. (2017) and Rosenfeld et al. (2019), it has been shown language modeling loss (i.e. cross-entropy) scales smoothly with model size and number of training tokens (Kaplan et al., 2020; Hoffmann et al., 2022). If we think of language models as essentially performing "imitation learning" on text, then a natural next question is

---

[1]Code: `https://github.com/princeton-nlp/il-scaling-in-games`

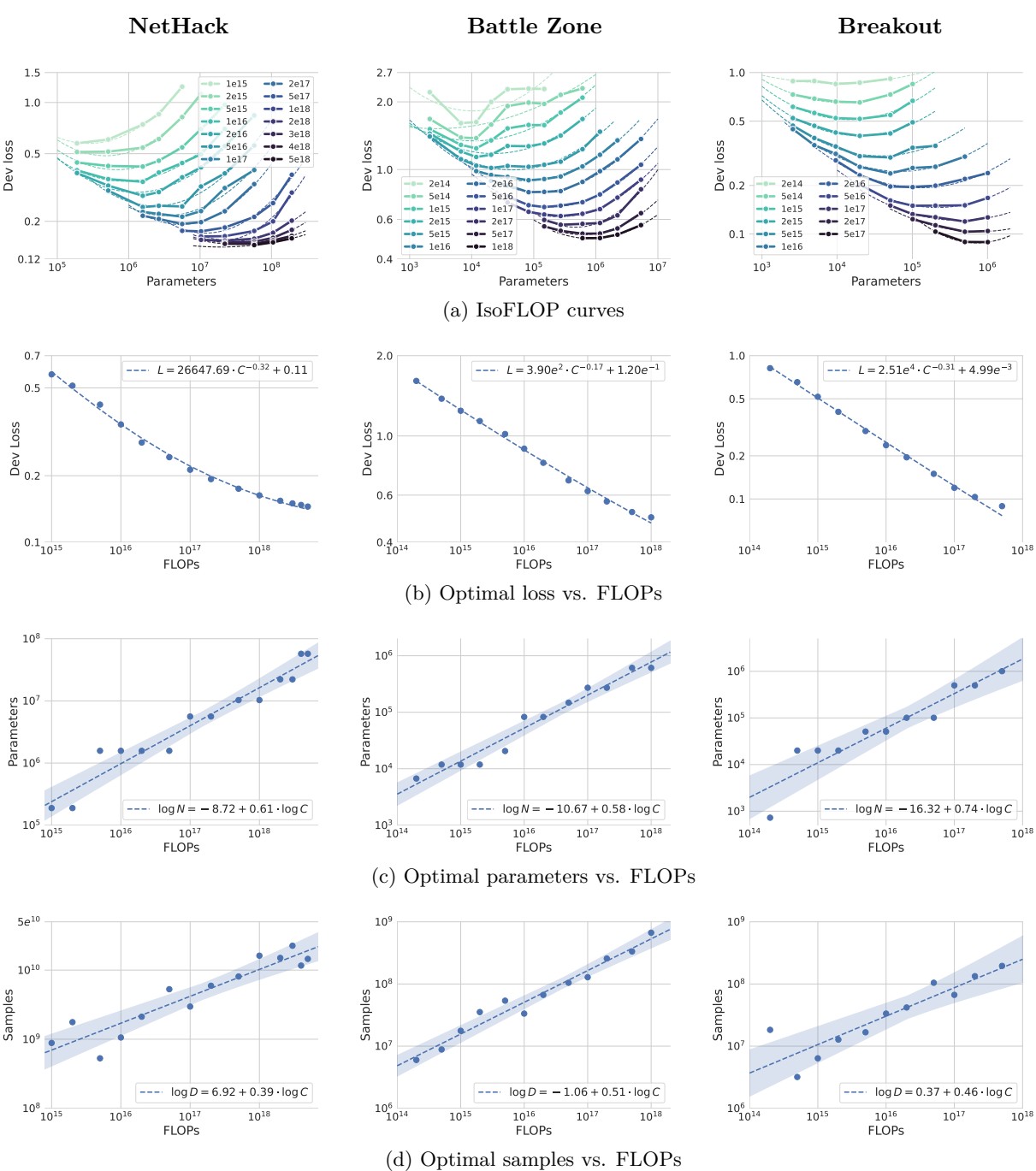

Figure 1: **BC loss scaling.** We train a wide range of model sizes across several orders of magnitudes of FLOP budgets. We plot the validation loss for each model, with fitted parabolas per IsoFLOPs curve (**a**). We then regress the loss minima (**b**), the loss-optimal number of parameters (**c**), and the loss-optimal number of samples (**d**) on their corresponding FLOP budgets. We find clear power law trends for Nethack (first column), Battle Zone (middle column), and Breakout (last column). The full list of Atari results can be found in Appendix J.

whether these results extend to IL-based agents in games, and whether scale could provide similar benefits and alleviate some of the issues mentioned earlier on.

In this paper, we ask the following question: *How does compute in terms of model and data size affect the performance of agents trained with imitation learning in the single-agent game setting?* We first focus on several Atari games with dense rewards which allows us to demonstrate our findings on a variety of games. However, since Atari games have all been solved at this point, there is not much room for further improvement in terms of scaling up. To demonstrate the potential of scaling up IL, our core focus will be on the extremely challenging game of NetHack, a roguelike video game released in 1987. NetHack is an especially well-suited and interesting domain to study for several reasons. First, it is procedurally generated and highly stochastic, disqualifying approaches relying heavily on memorization instead of generalization, such as Go-Explore (Ecoffet et al., 2021). Second, the game is partially observed, requiring the use of memory, potentially for thousands of steps due to the game's long-term dependencies. Finally, the game is extremely challenging for current AI systems, with current agents reaching scores nowhere close to average human performance[2]. The best agent on NetHack is a purely *rule-based* system called AutoAscend (Hambro et al., 2022a), with RL approaches lagging behind (Hambro et al., 2022b; Küttler et al., 2020; Mu et al., 2022; Mazoure et al., 2023). Even just recovering this system is hard, with Hambro et al. (2022b); Piterbarg et al. (2023a;b) all reporting that the best neural agents still *far* underperform the system's mean return in the environment, causing the authors to call for significant research advances. We instead investigate whether simply scaling up IL can help close some of this gap.

**Contributions.** We train a suite of neural Atari and NetHack agents with different model sizes using Behavioral Cloning (BC) to imitate expert policies and analyze the loss and mean return isoFLOP profiles. We find the optimal cross-entropy loss scales as a power law plus a constant in the compute budget, and we use two different methods to derive scaling laws for the loss-optimal model and data sizes. We then relate the cross-entropy loss of our trained BC agents to their respective mean return when rolled out in the environment, and find that the mean return follows an inverse power law plus constant with respect to the optimal cross-entropy loss, showing improvements in loss generally translate in better performing agents. We use our scaling law derivations to forecast the training requirements of several compute-optimal neural BC agent for NetHack. These forecasts are then verified to achieve the loss or mean return predicted by our scaling laws. Furthermore, our best agent outperforms prior neural NetHack agents by at least 1.7x in all settings, showing scale can provide dramatic improvements in performance for BC. We briefly extend our results to the RL setting, where we also train a suite of NetHack agents using IMPALA (Espeholt et al., 2018) and again find that model and data size scale as power laws in the compute budget. Finally, we also study the effect of partial observability on BC loss. Our results demonstrate that the improvements in imitation learning performance for single-agent games with dense rewards[3] can be described by power laws (and variations of them). This suggests carefully scaling up model and data size can provide big boosts to imitation learning performance in single-agent games, helping to narrow the gap between the learner and the expert.

## 2 Preliminaries

We now introduce the formal setup for behavioral cloning. We assume the environment can be described by a Partially Observable Markov Decision Process (POMDP) $\langle S, T, A, O, R, \gamma \rangle$, with states $S$, transition function $T$, action set $A$, possible observation emissions $O$, reward function $R(s, a)$, and discount factor $\gamma$.

In the behavioral cloning setup, we don't assume access to the rewards but instead assume access to a dataset $\mathcal{D}$ consisting of trajectories $\tau = (s_0, a_0, s_1, a_1, \ldots)$ of states and actions. These trajectories can be generated by multiple (possibly sub-optimal) demonstrators acting in the environment. However, in this work, they are assumed to all come from the same expert policy $\pi$. The goal is to recover this expert policy. To do this, a learner $\pi_\theta$ will optimize the following cross-entropy loss:

$$\mathcal{L}(\theta) = -\mathbb{E}_{(h_t, a_t) \sim \mathcal{D}} \left[ \log \pi_\theta(a_t | h_t) \right], \tag{1}$$

where $h_t$ can include part or the entirety of the history of past states and actions.

---

[2]The average overall human performance is around 127k (Hambro et al., 2022b), while the current best performing NetHack agent gets a score of 10k.

[3]Please refer to section 6 for a discussion of why we need this requirement.

## 3 Experimental setup

We analyze the scaling behavior of agents trained with BC in two domains: (1) Atari and (2) NetHack. The former serves to test the validity of the scaling laws in a range of games, while the latter tests the performance gains of scaling in an extremely challenging and unsolved game.

Whenever we report FLOP or parameter counts, we are referring to their *effective* counts, which we define as only including the parts of the network that are being scaled, similar to Hilton et al. (2023) (see Appendix E for full details). Please see Appendix G for details on all hyperparameters.

### 3.1 Atari

We chose the following set of 8 Atari games: Battle Zone, Q*bert, Name This Game, Phoenix, Space Invaders, Bank Heist, Boxing, and Breakout. For a detailed discussion on how these games were selected, please refer to Appendix F. We then perform the following steps for each game. First, we train a CNN-based agent with PPO (Schulman et al., 2017) in order to get an expert agent. Second, we gather a dataset of about 1B samples consisting of rollouts of the expert agent. We then train a family of CNN-based agents on this dataset using BC, varying the width of the core CNN and the final linear layer (see Appendix E). The total number of parameters ranged from 1k to 5M.

### 3.2 NetHack

We train Transformer-based agents on the NLD-AA dataset (Hambro et al., 2022b), varying both the width and depth (i.e. number of layers) of the model (see Appendix E). The total number of parameters ranged from 200k to 200M. While the original NLD-AA dataset already contains around 3B samples, we extended the dataset to around 55B samples (NLD-AA-L) by collecting more rollouts from AutoAscend (i.e. the data-generating policy).

## 4 Scaling up imitation learning

This section is structured as follows. We first investigate the role of model size and number of samples with respect to cross-entropy loss (subsection 4.1). While intuitively it feels like a lower loss should result in a better agent, we verify this by directly investigating the role of model size and number of samples with respect to the environment return (subsection 4.2), and relating these results to the loss results. Finally, we also show a possible extension of our analysis to the RL setting (subsection 4.3).

### 4.1 Scaling laws for BC loss

To investigate the role of model size and number of samples with respect to cross-entropy loss, we follow two approaches that are similar to the ones used in Hoffmann et al. (2022).

**Approach #1: isoFLOP profiles.** For Atari, we train up to 12 different model sizes, ranging from 1k to 5M. For NetHack, we train 10 different model sizes, ranging from 200k to 200M. For all domains, we train FLOP budgets as low as $1e14$ and up to $5e18$. In Figure 1 we plot the loss evaluated on a held-out set of about 100 (for Atari) and 10k (for NetHack) trajectories against the parameter count for each FLOP budget. Similarly to Hoffmann et al. (2022), we observe clear concave-up parabolas with well-defined minima at the optimal model size for a given compute budget in all games. We also find that loss decreases with increasing FLOP budgets. We take these loss-optimal data points to fit three regressions: one that regresses the parameters on the FLOPs, another that regresses the samples on the FLOPs, and a final one that regresses the loss on the FLOPs. These regressions give rise to the following power laws plus a constant:

$$N_{\text{opt}} = a_N C^\alpha + b_N \quad D_{\text{opt}} = a_D C^\beta + b_D \quad L_{\text{opt}} = a_L C^\gamma + b_L, \tag{2}$$

where $N_{\text{opt}}$ indicates the loss-optimal model size, $D_{\text{opt}}$ the loss-optimal number of training samples, $L_{\text{opt}}$ the minimal validation loss, and $C$ the compute budget in FLOPs. The $a_*$ and $b_*$ variables indicate additional

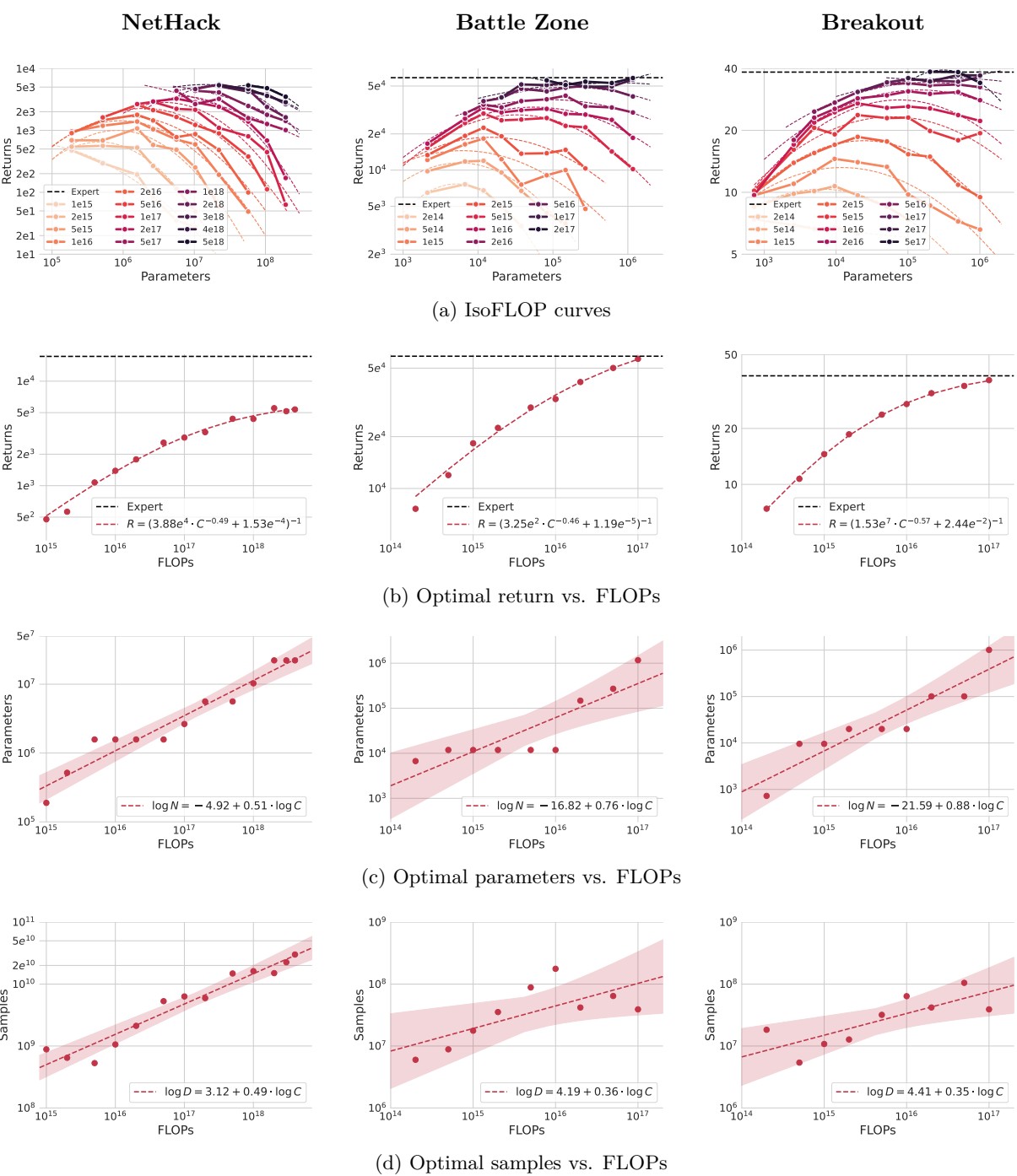

(a) IsoFLOP curves

(b) Optimal return vs. FLOPs

(c) Optimal parameters vs. FLOPs

(d) Optimal samples vs. FLOPs

Figure 2: **BC return scaling.** We train a wide range of model sizes across several orders of magnitudes of FLOP budgets (same models as in Figure 1a) and plot their average return in the environment (**a**). We then regress the optimal returns (**b**), the return-optimal number of parameters (**c**), and the return-optimal number of samples (**d**) on their corresponding FLOP budgets. We find mostly clear power law trends for Nethack (left), Battle Zone (middle), and Breakout (right). Full Atari results can be found in Appendix J.

fitted parameters. Note that sometimes we set the $b_*$ variables to 0 since they model the threshold of the power laws, which we don't always observe. For many of the loss power laws however, we do observe it and hence fit $b_L$ as well (e.g. see left plot of Figure 1b). We refer to the legends of Figure 1c, Figure 1d,

Table 1: **Fitted power law coefficients in NetHack.** We list the scaling coefficients for model size ($\alpha$) and number of samples ($\beta$) for all three settings. 95% CIs are noted in parentheses, where the delta method was used for the parametric fit parameters (see Appendix H).

| Setting | IsoFLOP profiles | | Parametric fit | |
|---|---|---|---|---|
| | $\alpha$ | $\beta$ | $\alpha$ | $\beta$ |
| 1. BC Loss | 0.61 (0.52, 0.71) | 0.39 (0.29, 0.48) | 0.590 (0.587, 0.593) | 0.410 (0.407, 0.413) |
| 2. BC Return | 0.51 (0.43, 0.59) | 0.49 (0.41, 0.57) | 0.605 (0.601, 0.610) | 0.395 (0.390, 0.399) |

and Figure 1b for sample values of the power law exponents $\alpha$, $\beta$, and $\gamma$, respectively. These figures indicate that as we increase FLOPs, we can expect loss to decrease according to a power law when appropriately increasing the size of the model and the size of the data according to their own power laws as well.

**Approach #2: parametric fit.** Instead of only fitting the loss-optimal points as was done in approach #1 above, one can also fit all points from Figure 1a to the following quadratic form:

$$\log \hat{L}(N, D) = \beta_0 + \beta_N \log N + \beta_D \log D$$
$$+ \beta_{N^2}(\log N)^2 + \beta_{ND} \log N \log D + \beta_{D^2}(\log D)^2. \tag{3}$$

If we only look at the linear terms here, we notice that this loss has the form of a Cobb-Douglas production function:

$$\hat{L}(N, D) = \exp(\beta_0) \times N^{\beta_N} \times D^{\beta_D}, \tag{4}$$

where we can think of parameters $N$ and samples $D$ as inputs that affect how much output (i.e. loss) gets produced. We then take the functional form in Equation 3 and minimize the loss subject to the constraint that FLOPs$(N, D) \approx 6ND^4$. To do this, we used the method of Lagrange multipliers to get the following functional forms for $N_{\text{opt}}$ and $D_{\text{opt}}$ (see Appendix A for full derivation):

$$N_{\text{opt}} = G\left(\frac{C}{6}\right)^{\alpha}, \quad D_{\text{opt}} = G^{-1}\left(\frac{C}{6}\right)^{\beta},$$
$$\text{where} \quad G = \exp\left(\frac{\beta_D - \beta_N}{2\beta_{D^2} - 2\beta_{ND} + 2\beta_{N^2}}\right). \tag{5}$$

We find that $\alpha = \frac{2\beta_{D^2} - \beta_{ND}}{2\beta_{D^2} - 2\beta_{ND} + 2\beta_{N^2}}$ and $\beta = \frac{2\beta_{N^2} - \beta_{ND}}{2\beta_{D^2} - 2\beta_{ND} + 2\beta_{N^2}}$. We compare the two approaches for NetHack in Table 1. We find both approaches to give similar scaling components for model and data size.

### 4.2 Scaling laws for BC return

Note that the analysis in the previous section was all in terms of cross-entropy loss. However, in the imitation learning setting, we almost never care directly about this quantity. Instead, we care about the average return of the resulting agent in the environment. To investigate how this quantity scales, we roll out every model from Figure 1a in the corresponding Atari or NetHack[5] environment and average their score across 100 rollouts in Atari, and across at least 500 rollouts in NetHack. The results in Figure 2a show we get a mirrored image of the loss case: we observe clear concave-down parabolas with well-defined maxima at the optimal model size for a given compute budget in all games. We also see that when FLOPs are increased, the returns

---

[4]Note that this FLOPs equation is only valid for our NetHack experiments, since the model there is Transformer-based. To carry out a similar analysis for Atari, where the models are CNN-based, this formula needs to be adjusted. We only perform the analysis for NetHack due to the simplicity of the FLOPs equation.

[5]While past work has pointed out the NetHack score is not necessarily aligned with winning the game (Küttler et al., 2020), they still recommend using it as a proxy to measure progress in the game.

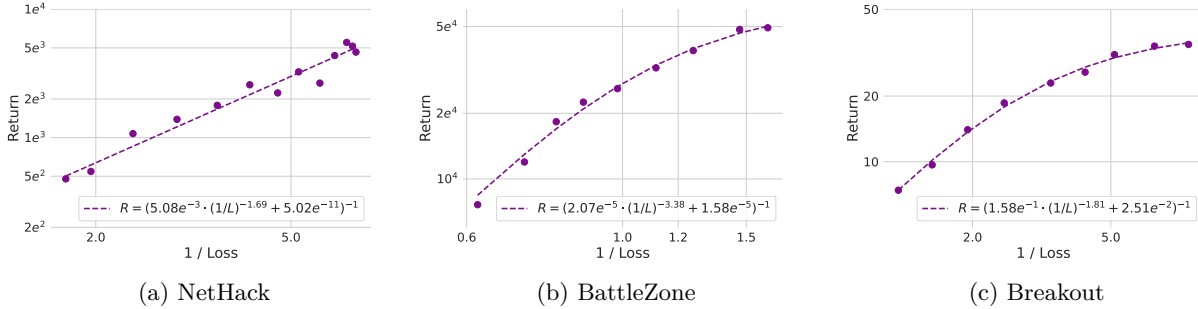

Figure 3: **BC return vs. optimal loss.** We investigate the relationship between the optimal loss of a BC agent and the mean return. We find they are highly correlated for all games.

go up. We then follow a similar procedure as in subsection 4.1 and perform the same three regressions, giving rise to the following power laws plus a constant (Figure 2c, Figure 2d, and Figure 2b):

$$N_{\text{opt}} = a_N C^\alpha + b_N \quad D_{\text{opt}} = a_D C^\beta + b_D \quad R_{\text{opt}} = (a_R C^\gamma + b_R)^{-1}, \tag{6}$$

where $N_{\text{opt}}$ indicates the return-optimal model size, $D_{\text{opt}}$ the return-optimal data size, $R_{\text{opt}}$ the maximal return, and $C$ the compute budget in FLOPs. We refer to the legends of Figure 2c, Figure 2d, and Figure 2b for sample values of $\alpha$, $\beta$, and $\gamma$, respectively. These figures indicate that as we increase FLOPs, the policy returns improve according to a power law when appropriately increasing the size of the model and the size of the data according to their own power laws as well. Note that the functional form for $R_{\text{opt}}$ is simply the inverse of a power law plus constant. When looking at Figure 2b, we find that for the Atari games the scaling laws hold all the way until expert performance. For NetHack, we find more FLOPs (and maybe other improvements, see the end of section 5 for a discussion on this) will be required to reach the expert score of ~17k.

Additionally, we can take the functional form in Equation 3 and simply replace loss with mean return. We can then solve the same constrained optimization problem resulting in the exact same expressions as found in Equation 5. We list the resulting coefficients for NetHack in Table 1. Unlike was the case for BC loss, we find the scaling exponents for the two methods to somewhat differ for BC return. While the parametric fit indicates scaling model and data size similarly to the loss case, the isoFLOP profiles indicate scaling model and data size equally instead. In general, we recommend performing a rolling cross-validation and picking the method with the lowest RMSE.

To investigate the relationship between loss and mean return, we regress the loss-optimal returns on the corresponding loss values. We find the relationship can be well described by the inverse of a power law plus a constant, i.e. $R = (a_{RL} (1/L_{\text{opt}})^\delta + b_{RL})^{-1}$, as shown in Figure 3. The fit in the figure shows optimal loss and mean return are highly correlated in all games, indicating we can expect return to increase smoothly as we make improvements in loss, rather than showing sudden jumps.

## 4.3 Extension to reinforcement learning

Given the stark trends we found for BC in the previous sections, we investigate whether similar trends can be found for RL. We explore this briefly for the game of NetHack since several works in the past years have attempted RL-based approaches for NetHack (Küttler et al., 2020; Hambro et al., 2022b) without coming close to solving the game, unlike is the case for Atari. We investigate the role of model size and environment interactions using approaches 1 and 2 from subsection 4.1 applied to IMPALA (Espeholt et al., 2018).

While learning curves in RL tend to have high variance, Figure 4 suggests that compute-optimal agents should increase both the number of parameters and number of environment interactions as the FLOP budgets are scaled up. We also find that the NetHack game score varies smoothly with FLOPs and hence can be seen as a *natural performance metric* (Hilton et al., 2023). We provide complete details of our setup and results in Appendix I.

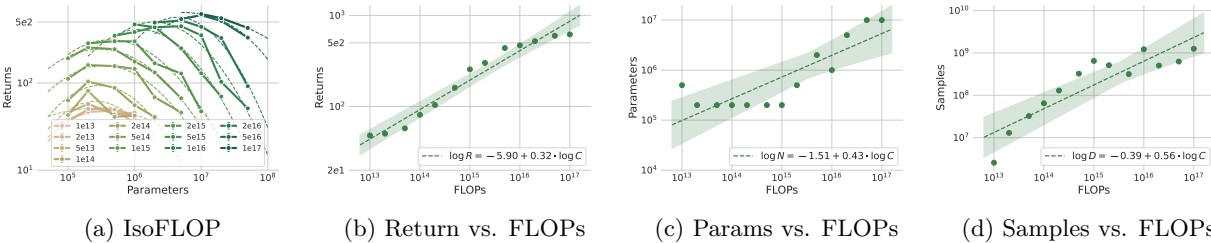

| (a) IsoFLOP | (b) Return vs. FLOPs | (c) Params vs. FLOPs | (d) Samples vs. FLOPs |

Figure 4: **RL return scaling.** We train a wide range of model sizes across several orders of magnitude of FLOP budgets and plot the average return when rolled out in the environment at the end of training (**a**). We then regress the return-optimal average returns (**b**), parameters (**c**), and samples (**d**) on their corresponding FLOP budgets. We train 1 seed per point on the isoFLOP profile.

## 5 Forecasting compute-optimal BC agents

The isoFLOP profiles and scaling laws shown in Figure 1 and Figure 2 allow us to predict the loss or return we can expect to achieve for a given compute budget, when allocating FLOPs optimally between model and data size. For all our Atari games (except for Space Invaders), we already trained agents that reached the expert return (see the black lines in Figure 2b), and we don't expect the agents to improve further with more compute, so these games are not very suitable to test our scaling laws for extrapolation. Hence, we will turn to NetHack to test the predictive power of our scaling laws, using the following extrapolation tests:

1. **Loss @ 40B**: Given 40B samples, predict the *loss-optimal* model size, and check if the resulting agent achieves a loss predicted by our loss scaling law.

2. **Return @ 40B & 55B**: Similar to the the loss prediction, we predict the *return-optimal* model size for both 40B samples and 55B samples. Then, we check if the resulting agents achieve a return predicted by our return scaling law.

While the predictions above start with a dataset size and calculate the optimal FLOPs and model size from there, one could just as well have started from a desired loss or return to achieve, a set model size, or a given FLOPs budget. Given any one of the aforementioned, one can compute any of the others (assuming we want to run in the compute optimal regime).

To compute the optimal model size for the predictions mentioned earlier, we use the scaling laws derived from the isoFLOP profiles[6] as follows. We first plug in $D = 40B$ into the regression in Figure 1d (for loss) or Figure 2d (for return) to solve for $C_{40B}$, the FLOP budget corresponding to a dataset size of 40B samples. Then, we plug $C_{40B}$ into the regression in Figure 1c (for loss) or Figure 2c (for return) to get $N_{40B}$, the compute-optimal model size when using 40B samples. This way, we find that the model size for the loss prediction @40B, the return prediction @40B, and the return prediction @55B should be 138M, 32M, and 45M, respectively.

We trained the three forecasting settings above which took ∼5 days for the @40B models, and ∼6.5 days for the @55B models. The results can be found in Figure 5. For the loss prediction, we find our scaling law matches the actual loss almost perfectly. Somewhat surprisingly, the return models performed a bit better than predicted by our scaling law. This could be due to using a cosine learning rate schedule for the predictions, while using a constant learning rate for the isoFLOP profiles from which the scaling laws are derived (see section 6 for a bit more discussion on this). In Table 2, we compare our best agent with offline RL methods (DQN-Offline, CQL, and IQL) as well as other past BC methods (CDGPT5, Transformer, and diff History LM). For completeness, we also list methods that use additional RL fine-tuning on top of offline

---

[6]We could have also used the parametric fit, and in practice we recommend performing a rolling cross-validation for both methods and choosing the method with the lowest RMSE.

Table 2: **Comparison with baselines.** We compare our best BC model with previous models in the `NetHackChallenge-v0` environment and find it outperforms all of them on the human monk role in the offline setting. *Exact scores not reported. See Appendix B for full results with standard errors.

| Models | Human Monk |
| --- | --- |
| **Offline only** | |
| DQN-Offline | 0.0 |
| CQL | 56 |
| IQL | 201 |
| BC (CDGPT5) | 1058 |
| BC (Transformer) | 1974 |
| diff History LM | 4504 |
| **Scaled-BC (ours)** | **7784** |
| **Offline + Online** | |
| Kickstarting + BC | 2090 |
| APPO + BC | 2809 |
| LDD* | 2100 |
| BC + Fine-tuning + KS | 10588 |
| **Dataset Average** | 17274 |

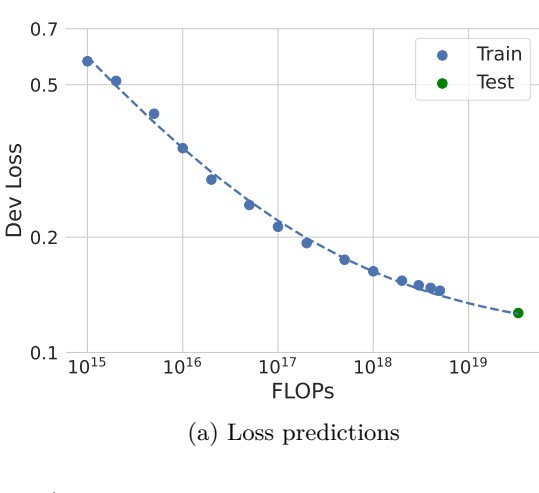

(a) Loss predictions

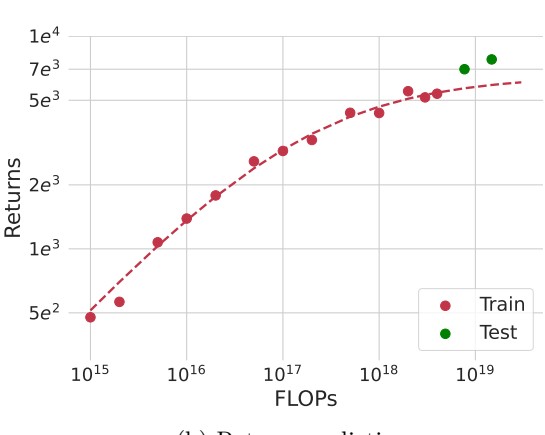

(b) Return predictions

Figure 5: **Forecasting Results.** In Table 2, we compare our models to prior work and find it sets a new SOTA in the *offline* setting. In (a), we show the loss-optimal model matches the prediction. In (b), we show the return-optimal models are close to our predictions.

training. Our agent gets a score of 7784 on the Human Monk role in NetHack - well beyond the next best offline method of 4504, which additionally relied on a pretrained initialization. This indicates BC performance can be boosted significantly with scale.

**Discussion**    While our return power law initially continues to improve with scale, we find it will plateau well before expert performance in NetHack (but not in Atari). There could be various reasons for this early plateauing of the power law, one of which is partial observability, which we study further in section 7.

## 6    Limitations

**Natural performance metrics.**    There is no reason in general to expect game scores to scale smoothly. If they do, Hilton et al. (2023) define them as *natural performance metrics*. Hence, one way of viewing our results is as a confirmation of the score functions for NetHack and Atari as natural performance metrics for IL. We expect that for any game score to be a natural performance metric, it needs to be at least somewhat dense so it tracks learning progress, which is why we focused on environments with relatively dense rewards in this paper[7]. It's possible our results extend to highly sparse reward settings as well, but one may need to introduce alternative proxy metrics (e.g. intrinsic performance (Hilton et al., 2023)) in that case.

---

[7]This, however, does *not* guarantee we will observe scaling laws in these environments when using IL!

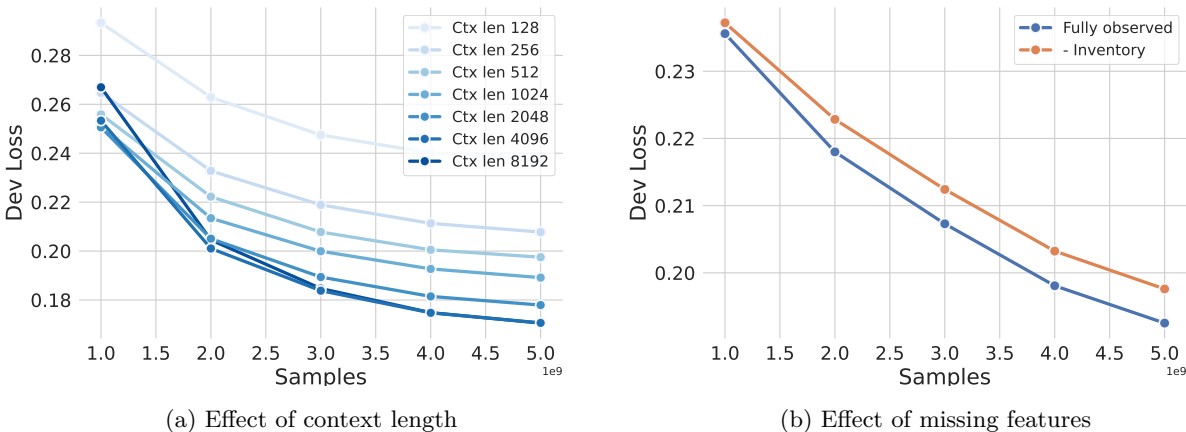

| (a) Effect of context length | (b) Effect of missing features |
|---|---|

Figure 6: **Effects of partial observability.** In (a) we observe context lengths up to 4096 give significant improvements in loss, suggesting long contexts may be needed to fully model the expert. In (b) we show that including the inventory, which has been commonly left out in past work but upon which the expert relies, further improves the loss.

**Experimental setup.** Previous works have pointed to the importance of tuning hyperparameters for every run on the isoFLOP profile. In particular, Hoffmann et al. (2022) recommend using a separate cosine learning rate schedule for every FLOP budget on the isoFLOP profile. However, to limit computational cost, we used a constant learning rate for every model size so we could leverage "snapshots" of the same run to evaluate different FLOP budgets for the same model size. While we did tune this constant learning rate pretty carefully (see Appendix G), there will nevertheless be some uncertainty in the exact values of all our power law coefficients. However, we expect the overall trends to still hold.

**Availability of data.** While some game environments might already come with large datasets of trajectories (Hambro et al., 2022b), this is generally not the case. Hence, depending on the game, the expert, and the scaling law, we might find ourselves in a data-constrained setting. If the game has a fast simulator and a computationally cheap expert (as is the case in this paper), then data availability may be less of a problem since we can simply collect more data by rolling out many trajectories in parallel using the expert policy. However, if the game simulator itself is slow or the expert is expensive (e.g. a human), then data availability may become a bottleneck. Finally, the specific scaling law for a game dictates how much data we actually need (assuming we want to run in the compute-optimal regime). Hence, if the scaling law indicates we'll need trillions of data points, we might be data-constrained somewhat irrespective of the computational requirements of running the game and the expert. The opposite is also true, however: for simple games (like Atari), the scaling laws seem to usually indicate we don't need that much data ($< 1B$), which means we might still be able to get away with slightly more computationally demanding game simulators and experts. One interesting direction for future work could be extending our results to the data-constrained setting, similar to Muennighoff et al. (2024).

**Comparing across environments and architectures.** While we find the *functional form* of the scaling laws to be the same across all environments (Atari and NetHack) and architectures (Transformer and CNN), we did not investigate the influence of different environment properties or architectures on the scaling coefficients. We leave such cross-comparisons of different environments and architectures to future work.

# 7 Partial observability

In this section, we discuss how partial observability can affect how well we can model the expert. Specifically, we focus on two dimensions: (1) context length, and (2) information parity.

**Context length.**  While Atari games are mostly Markovian, NetHack is highly partially observed, making the use of memory crucial. AutoAscend extensively keeps track of the past, making it important for our learner to also take into account the history. However, knowing exactly how much history is necessary to model AutoAscend is hard since it is a rule-based policy and hence it's not immediately clear what the "effective context length" of AutoAscend is. In Figure 6a, we study a wide range of context lengths and find that loss improves considerably up to lengths as long as 4096, after which improvements seem to be marginal. This suggests that past work trying to do BC on AutoAscend has either leveraged models that are known to struggle with modeling long contexts (Hambro et al., 2022a) (e.g. LSTMs) or simply is using context lengths that are too short (Piterbarg et al., 2023a;b).

**Information parity.**  We define *information parity* as the case when the learner has access to the same features of the environment as the expert does. Similarly as with context length, this is not relevant in the Atari games we study since the observation consists of a simple image of pixels, which both the learner and expert have full access to. In NetHack, however, the picture is murkier again because there is many observation features that NLE exposes of which AutoAscend makes extensive use, while prior work typically only uses a subset. In Figure 6b, we study leaving out the inventory features, which have been commonly left out in past work (Hambro et al., 2022a; Wolczyk et al., 2024) as they were not available in the original NLD dataset. We observe that including the inventory does in fact improve validation loss, suggesting that it is necessary to fully model the expert.

Our results on context length and information parity suggest that there may be hidden assumptions to our scaling laws, namely that they will be affected if the expert either keeps track of more history than the learner can, or is relying on features that the learner does not have access to. We leave a more extensive investigation of exactly *how* the scaling laws will be affected in the case of partial observability to future work.

## 8   Related work

**NetHack.**  Work on NetHack has been somewhat limited so far, with early work establishing the NLE benchmark (Küttler et al., 2020), evaluating symbolic vs. neural agents (Hambro et al., 2022a), and creating large-scale datasets based off of rule-based and human playthroughs for methods aiming to learn from demonstrations (Hambro et al., 2022b). Later work has either focused on better reward signal supervision and sample efficiency through proxy metrics and contrastive pre-training (Mazoure et al., 2023; Bruce et al., 2023) or leveraged dynamics models with language descriptions in order to improve sample efficiency and generalization (Mu et al., 2022). Piterbarg et al. (2023b) also investigates the gap between neural methods and AutoAscend, but focuses on leveraging an action hierarchy, improvements in architecture, and fine-tuning with RL. More recent work includes building long-context language agents (Piterbarg et al., 2023a) and investigating RL finetuning techniques to boost the performance of pretrained models (Wolczyk et al., 2024).

**Scaling laws.**  Hestness et al. (2017) and Rosenfeld et al. (2019) are one of the earliest works that try to characterize empirical scaling laws for deep learning. Kaplan et al. (2020) and Hoffmann et al. (2022) specifically focus on training compute-optimal language models, finding similar trends as presented in this paper. While in the imitation learning setting, our agents also minimize cross-entropy loss, we additionally show that the eventual performance of the agent as measured by the average return in the environment scales smoothly with the loss. Other works focus more broadly on generative modeling (Henighan et al., 2020), or analyze specific use cases such as acoustic modeling (Droppo & Elibol, 2021). Clark et al. (2022) investigate scaling laws for routing networks, and Hernandez et al. (2021) study scaling laws for transfer, finding the *effective data transferred* (the amount of extra data required to match a pre-trained model from scratch) follows a power-law in the low-data regime. More recent works have also tried to extend these scaling law results to multi-modal learning (Cherti et al., 2022; Aghajanyan et al., 2023). Caballero et al. (2022) introduce *broken neural scaling laws*, which allow modeling of double descent and sharp inflection points.

Perhaps the closest work to our paper is that of Hilton et al. (2023), who characterize scaling laws in RL. However, they don't consider IL, and they do not evaluate on Atari or NetHack, the latter of which we consider an especially interesting environment because of its challenging nature.

## 9    Discussion

**Beyond single-agent games.**   We have shown that in the imitation learning setting (and to some extend in the RL setting), scaling up model and data size provides predictable improvements, as demonstrated in a variety of Atari games and in the full game of NetHack. While we do not extend our analysis beyond single-agent games in this paper, we believe there are several key takeaways to take from our work when performing BC on a new domain. *First*, our results show that the same scaling laws show up across games with various properties (stochasticity, partial observability, pixel-based, etc.), suggesting they may show up for other domains as well. *Second*, poor performance (relative to the expert) in a new domain might be explained at least in part by scale. This is powerful since one might be able to stick with BC as a method instead of turning to alternatives as long as one is careful about the model and data size. *Third*, one has to be careful about partial observability. A gap in context length or information parity between the learner and the expert could potentially hurt BC performance.

**Leveraging human data.**   In this work, we did not consider analyzing the scaling relationships when using human trajectories (e.g. from NLD-NAO (Hambro et al., 2022b)) instead of those from AutoAscend (NLD-AA (Hambro et al., 2022b)). This is because extra care must be taken to handle the lack of actions in the human dataset, requiring techniques such as BCO (Torabi et al., 2018). Investigating scaling laws here could be especially interesting since: (1) the human dataset is more diverse, containing trajectories from many different players with varying level of skill, and (2) it contains many examples of trajectories that ascend (i.e. win the game). (1) could shed perspective on the role of scaling when the data includes many different and potentially suboptimal demonstrations, similar to Beliaev et al. (2022). (2) could provide insight into the viability of methods such as Video PreTraining (Baker et al., 2022) since these rely heavily on being able to clone the expert data well.

**The effect of expert quality.**   The experts used in our paper are of different qualities for different domains. For example, while AutoAscend is pretty poor compared to a human expert, the expert we use for Breakout is superhuman (Schwarzer et al., 2021). Nevertheless, we observe scaling laws in all cases, suggesting that the quality of the expert does not affect the *existence* of our scaling law. Second, what we do find to vary based on the quality of the expert is the upper bound or ceiling that the scaling laws converge to. Specifically, for all Atari games, we find that the scaling laws converge exactly to the mean return of the expert. Hence, we expect that as the expert quality improves or deteriorates for a particular environment, the corresponding scaling law will change to reflect this shift in the ceiling (i.e. it will now plateau at a different mean return).

## 10    Conclusion

In this work, we find that imitation learning loss and mean return follow clear scaling laws with respect to FLOPs, as demonstrated in Atari and in the challenging game of NetHack. In addition, we find loss and mean return to be highly correlated, meaning improvements in loss translate in improved performance in the environment. Using the found power laws, we forecast the compute requirements (in terms of model and data size) to train compute-optimal agents aimed at recovering the underlying expert. In NetHack, we find the performance improves substantially, surpassing prior SOTA by 1.7x in the offline setting. We also briefly extend our results to the reinforcement learning setting, and find similar power laws for model size and number of interactions in NetHack. Our results demonstrate that scaling up model and data size can provide big boosts to imitation learning performance in single-agent games. More broadly, they also call for work in the larger imitation learning and reinforcement learning community to more carefully consider and study the role of scaling laws, which could provide large improvements in many other domains.

## Broader Impact Statement

While we do not see a direct path towards any negative applications, we note that scaling up could have unforeseen consequences. As scaling results in imitation and reinforcement learning agents that are increasingly more capable and influential in our lives, it will be important to keep them aligned with human values.

## Acknowledgements

We thank Alexander Wettig, Ameet Deshpande, Dan Friedman, Howard Chen, Jane Pan, Mengzhou Xia, Khanh Nguyen, Shunyu Yao, and Vishvak Murahari from the Princeton NLP group for valuable feedback, comments, and discussions. We are also grateful to Riccardo Savorgnan, Sohrab Andaz, Tessa Childers-Day, Carson Eisenach, Kenny Shirley, and others from the Amazon SCOT Forecasting team for helpful discussions and encouragement. We thank Kurtland Chua for helpful feedback. Finally, we give special thanks to Eric Hambro for answering any questions we had about the NetHack environment and datasets throughout the project. Sham Kakade acknowledges funding from the Office of Naval Research under award N00014-22-1-2377 and the National Science Foundation Grant under award #CCF-2212841. JT and KN acknowledge support from the National Science Foundation under Grant No. 2107048. Any opinions, findings, and conclusions or recommendations expressed in this material are those of the author(s) and do not necessarily reflect the views of the National Science Foundation.

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

## A    Parametric fit: derivation of power law

In this section, we show the derivation resulting in Equation 5. We first restate Equation 3 for convenience:

$$\log \hat{L}(N, D) = \beta_0 + \beta_N \log N + \beta_D \log D$$
$$+ \beta_{N^2} \log^2 N + \beta_{ND} \log N \log D + \beta_{D^2} \log^2 D.$$

Now, to minimize this equation with the constraint that $\text{FLOPs}(N, D) = C \approx 6ND$, we use Lagrange multipliers. We first write down the Lagrangian:

$$\mathcal{L}(N, D, \lambda) = \hat{L}(N, D) - \lambda(g(N, D) - C),$$

where $g(N, D) = 6ND$. Now, setting $\nabla \mathcal{L} = \mathbf{0}$ we get:

$$\nabla \log \hat{L}(N, D) = \lambda \nabla (6ND) \quad \text{and} \quad 6ND = C.$$

The former results in the following system of equations:

$$\frac{\partial \log \hat{L}(N, D)}{\partial N} = \lambda \frac{\partial (6ND)}{\partial N}$$
$$\frac{\partial \log \hat{L}(N, D)}{\partial D} = \lambda \frac{\partial (6ND)}{\partial D}.$$

This means that

$$\beta_N \frac{1}{N} + \beta_{ND} \log D \frac{1}{N} + 2\beta_{N^2} \log N \frac{1}{N} = \lambda 6D$$
$$\beta_D \frac{1}{D} + \beta_{ND} \log N \frac{1}{D} + 2\beta_{D^2} \log D \frac{1}{D} = \lambda 6N.$$

Multiplying the top equation by $N$ and the bottom one by $D$ we have that

$$\beta_N + \beta_{ND} \log D + 2\beta_{N^2} \log N = \beta_D + \beta_{ND} \log N$$
$$+ 2\beta_{D^2} \log D.$$

Recalling $C = 6ND$, we solve for $N$ and $D$ in terms of $C$, giving the results listed in Equation 5.

## B    Full set of forecasting results for NetHack

The full set of forecasting results for NetHack can be found in Table 3. DQN-Offline, CQL, and IQL are all offline RL methods, while BC (CDGPT5), BC (Transformer), and diff History LM are all based on behavioral cloning.

## C    Full set of isoFLOP figures for NetHack

We show the full set of isoFLOP curves in NetHack for all settings in Figure 7.

## D    Architecture details

### D.1    Atari

Our architecture for BC experiments in Atari is simple. It consists of the following layers:

1. A convolutional layer with 8 x 8 filter size and stride 4, followed by a ReLU activation layer.

2. A convolutional layer with 4 x 4 filter size and stride 2, followed by a ReLU activation layer.

3. A convolutional layer with 3 x 3 filter size and stride 1, followed by a ReLU activation layer.

4. A final linear layer that maps the flattened output dimension of the CNN layers above to the number of actions in the respective Atari game.

Table 3: **Forecasting results (full).** Table 2 with added standard errors. Results from Hambro et al. (2022b) use 10 seeds, while those from Piterbarg et al. (2023b) and ours are 1 seed. *Exact scores not reported in original work.

|  | Human Monk |
|---|---|
| **Offline only** | |
| DQN-Offline (Hambro et al., 2022b) | $0.0 \pm 0.0$ |
| CQL (Hambro et al., 2022b) | $56 \pm 28$ |
| IQL (Hambro et al., 2022b) | $201 \pm 27$ |
| BC (CDGPT5) (Hambro et al., 2022b;a) | $1058 \pm 159$ |
| BC (Transformer) (Piterbarg et al., 2023b) | $1974 \pm$ - |
| diff History LM (Piterbarg et al., 2023a) | $4504 \pm$ - |
| **Scaled-BC (ours)** | **$7784 \pm$ -** |
| **Offline + Online** | |
| Kickstarting + BC (Hambro et al., 2022b) | $2090 \pm 123$ |
| APPO + BC (Hambro et al., 2022b) | $2809 \pm 102$ |
| LDD* (Mu et al., 2022) | $2100 \pm$ - |
| BC + Fine-tuning + KS (Wolczyk et al., 2024) | $10588 \pm 672$ |

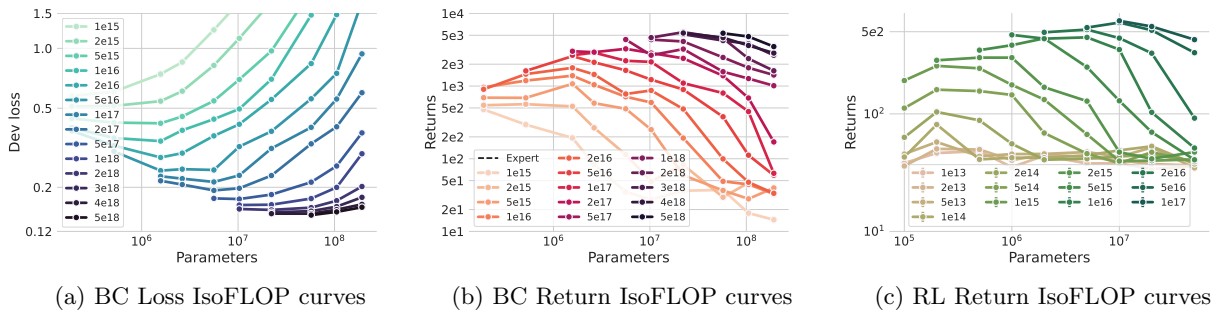

(a) BC Loss IsoFLOP curves    (b) BC Return IsoFLOP curves    (c) RL Return IsoFLOP curves

Figure 7: **Full isoFLOP curves.** (**a**) is similar to Figure 1a but including the full set of points. (**b**) is similar to Figure 2a but including the full set of points. (**c**) is similar to Figure 4a but including the full set of points.

### D.2    NetHack

We use two main architectures for all our experiments, one for the BC experiments and another for the RL experiments.

**BC architecture.**    The NLD-AA dataset (Hambro et al., 2022b) is comprised of *ttyrec*-formatted trajectories, which are $24 \times 80$ ASCII character and color grids (one for each) along with the cursor position. To further reduce partial observability, we regenerate the dataset to also store the dungeon and inventory glyphs. To encode the ASCII characters and colors along with the glyphs, we modify the architecture used in Hambro et al. (2022b), resulting in the following:

- **Dungeon encoder.** This component encodes the main observation in the game, which is a $21 \times 80$ grid per time step. Note the top row and bottom two rows are cut off as those are fed into the message and bottom line statistics encoder, respectively. We embed each character, color, and glyph in an embedding lookup table, concatenate them together, and feed them through one linear layer, after which we put them in their respective positions in the grid. We then feed this embedded grid into a ResNet, which consists of two identical modules, each using one convolutional layer followed

by a max pooling layer and two residual blocks (of two convolutional layers each), for a total of 10 convolutional layers, closely following the setup in Espeholt et al. (2018).

- **Message encoder.** The message encoder takes the top row of the grid, converts all ASCII characters into a one-hot vector, and concatenates these, resulting in a $80 \times 256 = 20{,}480$ dimensional vector representing the message. This vector is then fed into a two-layer MLP, resulting in the message representation.

- **Bottom line statistics.** To encode the bottom line statistics, we flatten the bottom two rows of the grid and create a "character-normalized" (subtract 32 and divide by 96) and "digits-normalized" (subtract 47 and divide by 10, mask out ASCII characters smaller than 45 or larger than 58) input representation, which we then stack, resulting in a $160 \times 2$ dimensional input. This closely follows the Sample Factory[8] model used in Hambro et al. (2022b).

- **Inventory.** Unlike prior work (Hambro et al., 2022a), we also include the inventory glyphs in our network to increase information parity with respect to the expert (see section 7). We use the same glyph embedding table as used for the dungeon encoder followed by a linear projection. Then, we concatenate all inventory items along the hidden dimension and feed them through a two-layer MLP. Note that we only use these inventory features for IL, not for RL.

After the components above are encoded, we concatenate all of them together. Additionally, we also concatenate the previous frame's action representation (coming from an embedding lookup table), and a crop representation (a $9 \times 9$ crop around the player, processed by a five-layer CNN). We then feed this combined representation into a two-layer MLP, after which a Transformer (in case of IL) or LSTM (in case of RL) processes the representation further. Finally, we have two linear heads on top of the sequence model, one for the policy and one for the value (not used for BC).

**RL architecture.** We modify the architecture from Küttler et al. (2020) to also include a five-layer one-dimensional CNN that processes the message, as well as another five-layer two-dimensional CNN that processes a $9 \times 9$ crop of the dungeon grid around the player.

# E  FLOP and parameter counting

As mentioned in the main text, we only count FLOPs and parameters for the parts of the model being scaled.

## E.1  Atari

The model network consists of three convolutional layers interleaved with ReLUs (see Appendix D for details), followed by a linear layer (i.e. the policy head). We simply scale the width of all layers. This means we scale the channels of the convolutional network and the width of the linear head on top.

Since we scale the whole network, we count the FLOPs from all convolutional and linear layers. We compute the forward FLOPs of a convolutional layer as $2 \cdot h_{\text{out}} \cdot w_{\text{out}} \cdot c_{\text{out}} \cdot p$, where $h_{\text{out}}$ is the height of the output shape, $w_{\text{out}}$ is the width of the output shape, $c_{\text{out}}$ are the number of output channels, and $p$ indicates the number of parameters in one filter of the current layer (without counting bias). Hence, $p = k^2 \cdot c_{\text{in}}$, where $k$ is the kernel size and $c_{\text{in}}$ is the number of input channels. Following prior work (Hilton et al., 2023), we assume the backward pass takes about twice the number of FLOPs from the forward pass.

## E.2  NetHack

For all our BC and RL experiments, we only scale the following parts of the model and keep the rest fixed:

- The hidden size of the two-layer MLP right before the Transformer (in the case of BC) or the LSTM (in the case of RL).

---

[8]https://github.com/Miffyli/nle-sample-factory-baseline

- The hidden size of the Transformer (in the case of BC) and the LSTM (in the case of RL).

- The input size of the two linear layers for the actor and critic respectively (i.e. the policy and value heads).

Similar to prior work (Kaplan et al., 2020; Hoffmann et al., 2022), we found $6ND$ to be a good approximation (for both the Transformer and the LSTM) for the number of FLOPs used based on model size $N$ and number of samples $D$. This is because we found there to be about $2ND$ FLOPs in the forward pass, and we assume the backward pass takes about the twice the number of FLOPs from the forward pass.

For the RL experiments, there is a slight change in the way we count FLOPs, which is that we count every forward pass number of FLOPs from the learner twice, since there is a corresponding forward pass from an actor. Hence, for RL our formula becomes $8ND$.

## F  Game Selection for Atari Games

We chose the following set of 8 Atari games: Battle Zone, Q*bert, Name This Game, Phoenix, Space Invaders, Bank Heist, Boxing, and Breakout. The first 4 games were picked from the Atari-5 subset (Aitchison et al., 2023), which tries to condense the full set of Atari games to a subset of 5 representative games. We had some trouble training a good (i.e. substantially better than random) expert for Double Dunk (the 5th of the Atari-5 subset), so instead we added 4 other games, chosen at random from the full set of Atari games (though excluding Montezuma's Revenge due to its extremely sparse rewards). We didn't include more than 8 games due to constraints on both compute and experimenter time, but we strongly suspect our results will generalize to most games in the Atari set. Note that the results could even hold true for Montezuma's revenge as well, but since we didn't specifically seek out very sparse reward games, we do not make that claim and instead convey this as a potential limitation.

## G  Training details

### G.1  Atari

We list the hyperparameters for all our BC experiments in Atari in Table 4a and the hyperparameters to train expert policies for each Atari game in Table 4b. To train these expert policies, we used the Stable Baselines3 (Raffin et al., 2021) implementation of PPO (Schulman et al., 2017). We use Adam (Kingma & Ba, 2014) as our optimizer for both settings. All training experiments were done on NVIDIA GPUs (a mix of GeForce RTX 3090, GeForce RTX 2080 Ti, RTX A5000, and RTX A6000) and took about 1 - 2 days depending on the game and FLOP budget.

Table 4: **Hyperparameters for all experiments in Atari.** We list the hyperparameters for all our BC experiments (**a**) as well as the ones used to train the PPO expert agent for each game (**b**).

| Hyperparameters | Value |
| --- | --- |
| Learning rate | 0.0001 |
| Batch size | $128 \times 1$ (1 GPU) |
| Unroll length | 32 |
| Data workers | 30 |

(a) BC hyperparameters

| Hyperparameter | Value |
| --- | --- |
| Learning rate | $2.5e^{-4}$ |
| Learner batch size | 256 |
| Entropy cost | 0.01 |
| Baseline cost | 0.5 |
| Total timesteps | $5e^7$ |
| Discount factor | 0.99 |
| GAE lambda | 0.95 |
| Clip range | 0.2 |

(b) PPO hyperparameters

### G.2 NetHack

We use AdamW (Loshchilov & Hutter, 2019) as our optimizer for all BC experiments. For RL, we use RMSprop. Please find all hyperparameters for both settings in Table 5, all of which were manually tuned or taken from prior work. All NetHack BC experiments were run on NVIDIA H100 80GB GPUs. All Atari BC experiments were run on a mixture of NVIDIA A5000 and A6000 GPUs. The RL experiments were run on V100 32GB GPUs. All experiments took anywhere from a few hours to 4 days to run. The one exception to this is the forecasted BC models which took up to 6.5 days to run.

The NLD-AA dataset (Hambro et al., 2022b) is released under the NetHack General Public License and can be found at `https://github.com/dungeonsdatasubmission/dungeonsdata-neurips2022`.

In Figure 8a, we investigate the optimal learning rate for both cosine and constant learning rate schedules in NetHack. We find 0.0005 works best for the constant schedule and hence use this for generating the isoFLOP profiles in the paper. For the predictions, we use a cosine schedule since Figure 8a shows it works better. In addition, in Figure 8b we test width vs. depth scaling and find that keeping the aspect ratio (i.e. hidden dimension divided by the number of layers) is best kept at a little over 100, which we stick to when generating our isoFLOP profiles.

Table 5: **Hyperparameters for all experiments in NetHack.** We list the hyperparameters for all our BC experiments (**a**) as well as the ones for our RL experiments (**b**). For BC, we always use a constant learning rate except for the predictions in section 5, for which we use a cosine schedule with warmup.

| Hyperparameters | Value |
|---|---|
| Learning rate | 0.0005 |
| Batch size | 32 |
| Unroll length | 8192 |
| Ttyrec workers | 30 |

(a) BC hyperparameters

| Hyperparameter | Value |
|---|---|
| Learning rate | 0.0002 |
| Learner batch size | 32 |
| Unroll length | 80 |
| Entropy cost | 0.001 |
| Baseline cost | 0.5 |
| Maximum episode steps | 5000 |
| Reward normalization | yes |
| Reward clipping | none |
| Number of actors | 90 |
| Discount factor | 0.99 |

(b) IMPALA hyperparameters

## H  Derivation of delta method

Following standard linear regression assumptions, we have that $\sqrt{n}(\hat{\boldsymbol{\beta}} - \boldsymbol{\beta}) \xrightarrow{D} \mathcal{N}(0, \Sigma)$, where $\hat{\boldsymbol{\beta}}$ is a vector containing all regression coefficients, i.e.

$$\hat{\boldsymbol{\beta}} = \langle \hat{\beta}_0, \hat{\beta}_N, \hat{\beta}_D, \hat{\beta}_{N^2}, \hat{\beta}_{ND}, \hat{\beta}_{D^2} \rangle$$

We also rewrite $\alpha$ from Equation 5 more explicitly as a function $h$:

$$\alpha = h(\hat{\boldsymbol{\beta}}) = \frac{2\beta_{D^2} - \beta_{ND}}{2\beta_{D^2} - 2\beta_{ND} + 2\beta_{N^2}}$$

Approximating $h(\hat{\boldsymbol{\beta}})$ with a first-order Taylor expansion around $\boldsymbol{\beta}$ we get:

$$h(\hat{\boldsymbol{\beta}}) \approx h(\boldsymbol{\beta}) + \nabla h(\boldsymbol{\beta})^T \cdot (\hat{\boldsymbol{\beta}} - \boldsymbol{\beta})$$

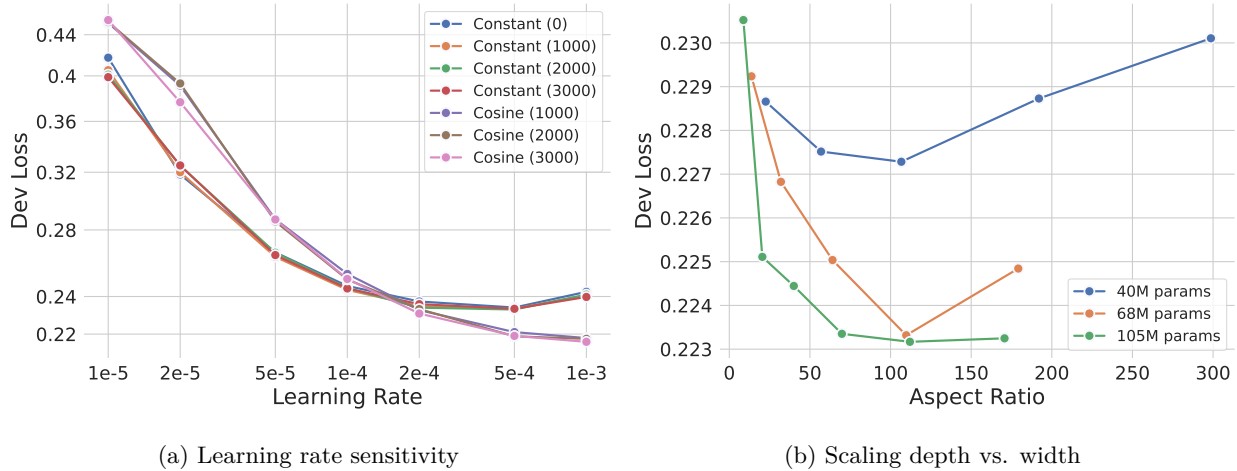

(a) Learning rate sensitivity

(b) Scaling depth vs. width

Figure 8: **IsoFLOP learning rate & aspect ratio in NetHack.** In **(a)**, we test both cosine and constant learning rate schedules with varying warmup periods (specified in number of gradient steps). We find 0.0005 to work best when using a constant learning rate. In **(b)**, we investigate the optimal aspect ratio and find it remains constant across model sizes.

Computing the variance of $h(\hat{\boldsymbol{\beta}})$ then gives[9]:

$$\begin{aligned}
\mathrm{Var}[h(\hat{\boldsymbol{\beta}})] &= \mathrm{Var}[h(\boldsymbol{\beta}) + \nabla h(\boldsymbol{\beta})^T \cdot (\hat{\boldsymbol{\beta}} - \boldsymbol{\beta})] \\
&= \mathrm{Var}[h(\boldsymbol{\beta}) + \nabla h(\boldsymbol{\beta})^T \cdot \hat{\boldsymbol{\beta}} - \nabla h(\boldsymbol{\beta})^T \cdot \boldsymbol{\beta}] \\
&= \mathrm{Var}[\nabla h(\boldsymbol{\beta})^T \cdot \hat{\boldsymbol{\beta}}] \\
&= \nabla h(\boldsymbol{\beta})^T \cdot \mathrm{Cov}(\hat{\boldsymbol{\beta}}) \cdot \nabla h(\boldsymbol{\beta}) \\
&= \nabla h(\boldsymbol{\beta})^T \frac{\Sigma}{n} \nabla h(\boldsymbol{\beta})
\end{aligned}$$

Now the Standard Error (SE) can be found as

$$\mathrm{SE} = \sqrt{\nabla h(\boldsymbol{\beta})^T \frac{\Sigma}{n} \nabla h(\boldsymbol{\beta})}.$$

Finally, we approximate the SE above by using $\nabla h(\hat{\boldsymbol{\beta}})$ instead of $\nabla h(\boldsymbol{\beta})$, giving the following confidence interval:

$$\mathrm{CI} = h(\hat{\boldsymbol{\beta}}) \pm 1.96 \cdot \mathrm{SE}.$$

## I    Scaling laws for reinforcement learning in NetHack

Table 6: Model and data size predictions for RL scaling laws in NetHack.

| *RL - Avg. Human (127k)* | | |
|---|---|---|
| 1. IsoFLOP profiles | 4.4B | 13.2T |
| 2. Parametric fit | 67B | 0.93T |

We train LSTM-based agents on the `NetHackScore-v0` environment. `NetHackScore-v0` features the full game of NetHack but has a reduced action space, starting character fixed to human monk, and automatic

---

[9]We follow a very similar derivation as in `https://en.wikipedia.org/wiki/Delta_method#Multivariate_delta_method`

menu skipping. We also experimented with the `NetHackChallenge-v0` environment but found the results too noisy at the FLOP budgets we were able to run. However, we expect similar results will hold for this environment at larger FLOP budgets.

**IsoFLOP profiles.** We train 9 different model sizes ranging from 100k to 50M using IMPALA (Espeholt et al., 2018), each with a FLOP budget ranging from $1e13$ to $1e17$. For each of these models, we evaluate the model at the end of training by rolling it out 1k times in the environment and reporting the average return. While learning curves in RL tend to have high variance, we generally still find that compute-optimal models should increase both the number of parameters and number of environment interactions as the FLOP budgets are scaled up (see Figure 4). We also find that the NetHack game score varies smoothly with FLOPs and hence can be seen as a *natural performance metric* (Hilton et al., 2023), as discussed in more detail in section 6. We again follow a similar procedure as in subsection 4.1 resulting in power laws as listed in Equation 6. We find $\alpha = 0.43$, $\beta = 0.56$, and $\gamma = 0.32$.

**Parametric fit** We take the functional form in Equation 3, and replace loss with mean return. We can then solve the same constrained optimization problem resulting in the exact same expressions as found in Equation 5 (the denominator of 6 is replaced with 8 due to a slight difference in FLOP counting for RL, see Appendix E). After fitting, we find $\alpha = 0.6$ and $\beta = 0.4$. Note we dropped the low flop budgets when performing this regression, as we found this greatly improved the fit.

**Forecasting human performance.** Hambro et al. (2022b) report that average human performance is around 127k. Based on the two approaches discussed above, we forecast the compute requirements for training an RL agent from scratch to achieve human-level performance on NetHack, listed in Table 6. For the isoFLOP profile approach, we first use Figure 4b to solve for $C_{127k}$. Then we plug this into the power laws from Figure 4c and Figure 4d. For the parametric fit, we instead plug $C_{127k}$ into the power laws from Equation 5 with the correct $\alpha$ and $\beta$ from above, where the denominator of 6 is replaced with 8 as mentioned earlier. In Table 6, we find the parametric fit to put significantly more emphasis on model size, which could be possible due to dropping of the low FLOP budgets (optimal model size tends to shift more clearly in larger FLOP budgets). Due to computational constraints, we leave testing the limits of this prediction to future work.

**RL with pretraining.** All our scaling law results on the RL side in this paper are with policies trained *from scratch*. However, some of the most promising neural methods for NetHack and other domains leverage a pre-trained (e.g. through imitation learning) policy that is then finetuned with RL. It would be very interesting to analyze the scaling behaviors for these kind of kickstarted policies, and see whether they scale differently than the ones trained from scratch. We leave this to future work.

## J Full results for Atari

 Figure 9, Figure 10, Figure 11, and  Figure 12 list the full set of Atari results with respect to cross entropy loss. Figure 13, Figure 14, Figure 15, and Figure 16 list the full set of Atari results with respect to environment return. Finally, Figure 17 lists the full set of Atari results relating environment return and optimal loss. Note that for the return results, we can see that Space Invaders is the only Atari game where didn't run high enough FLOP budgets to reach expert performance.

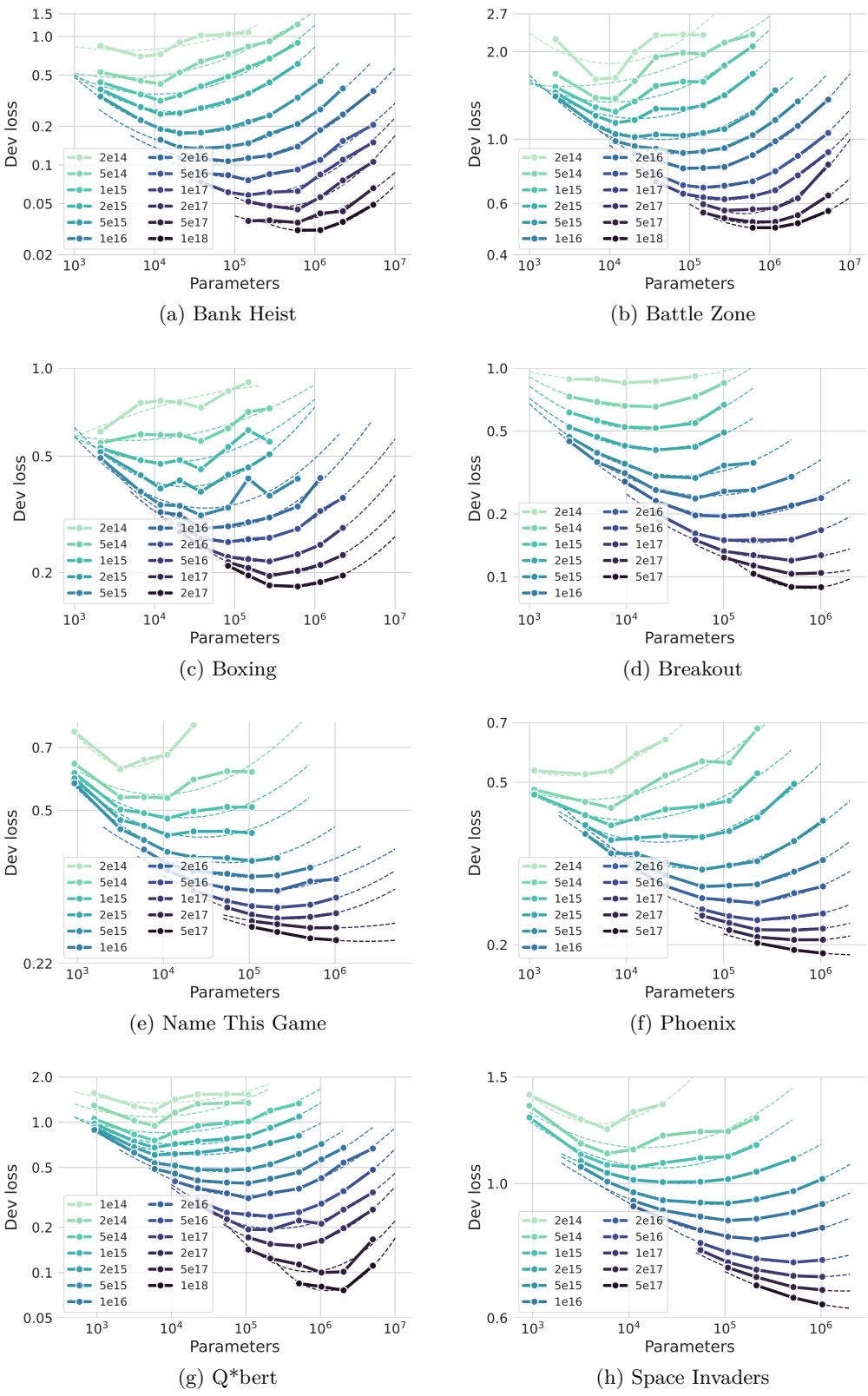

Figure 9: **BC isoFLOP curves.** Full results for all Atari games.

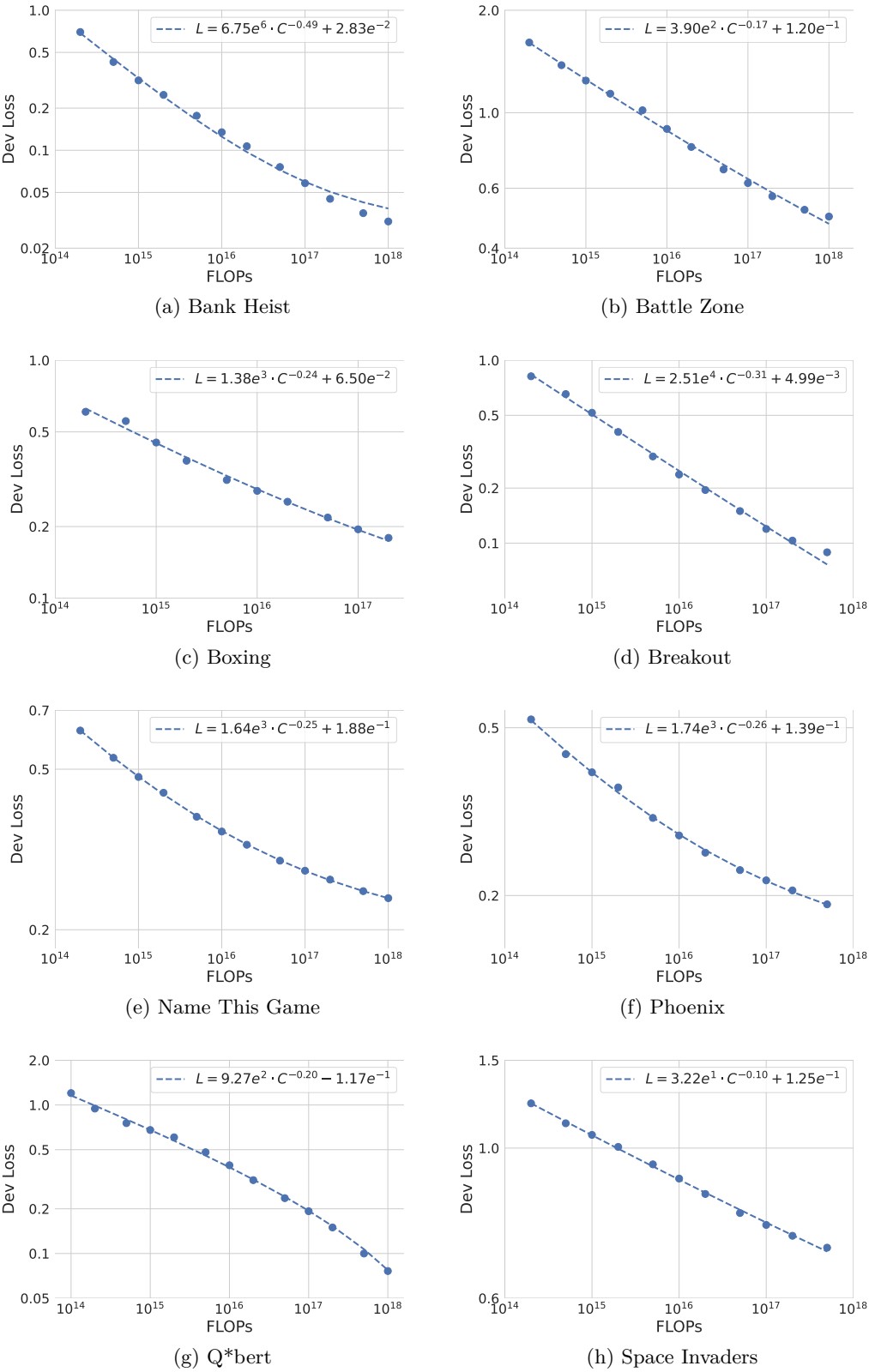

Figure 10: **BC optimal loss vs. FLOPs curves.** Full results for all Atari games.

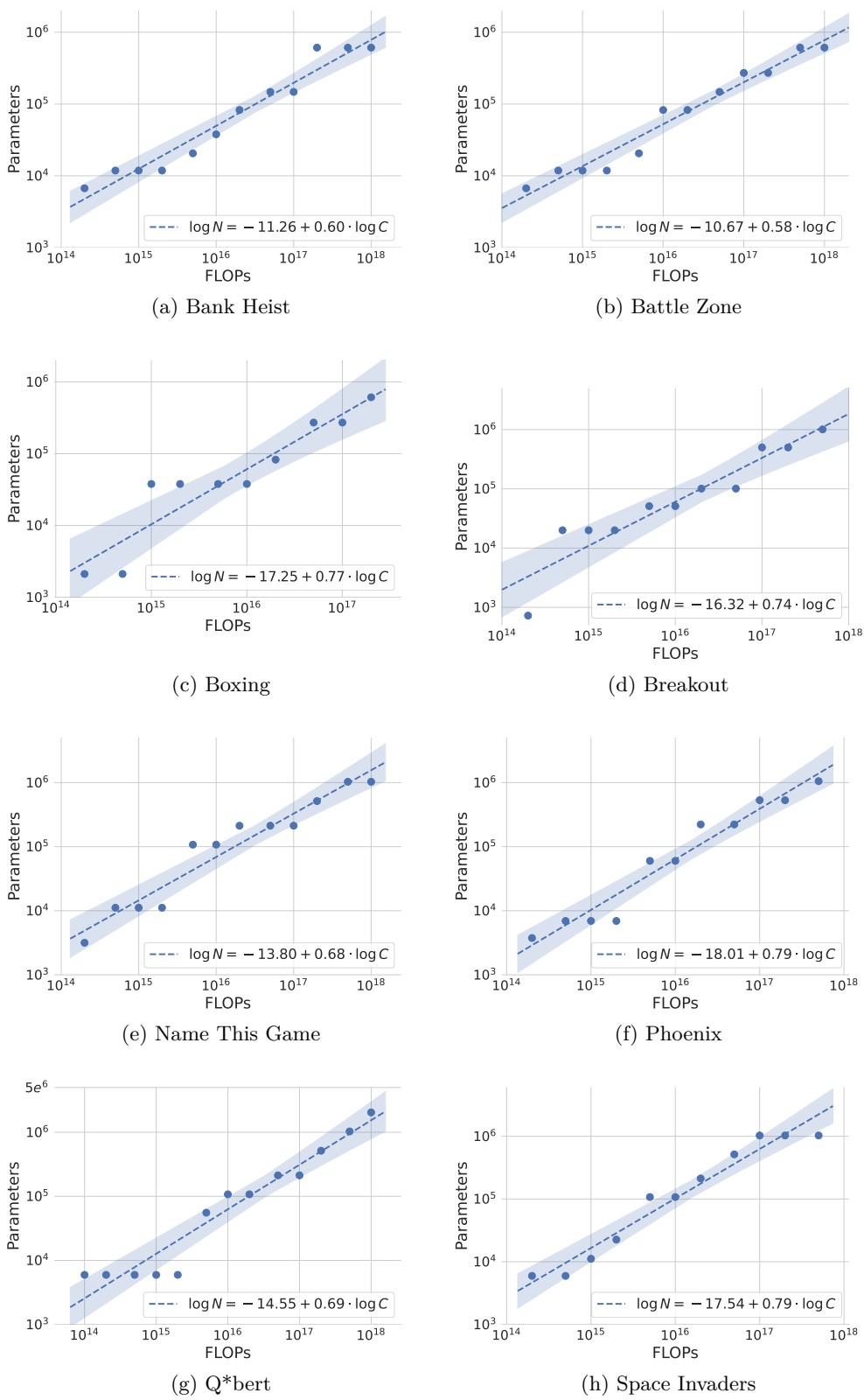

Figure 11: **BC optimal parameters vs. FLOPs curves.** Full results for all Atari games.

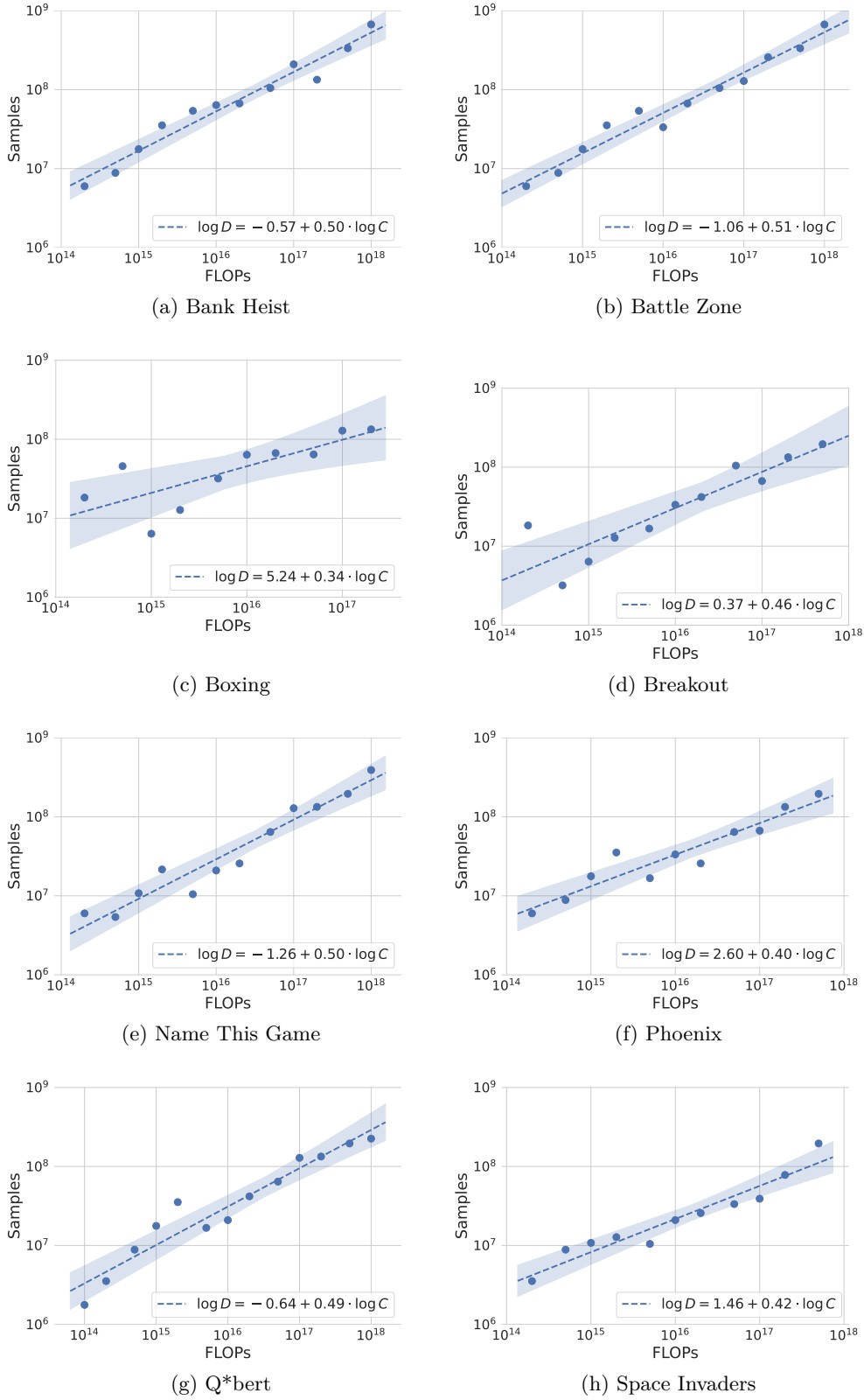

Figure 12: **BC optimal samples vs. FLOPs curves.** Full results for all Atari games.

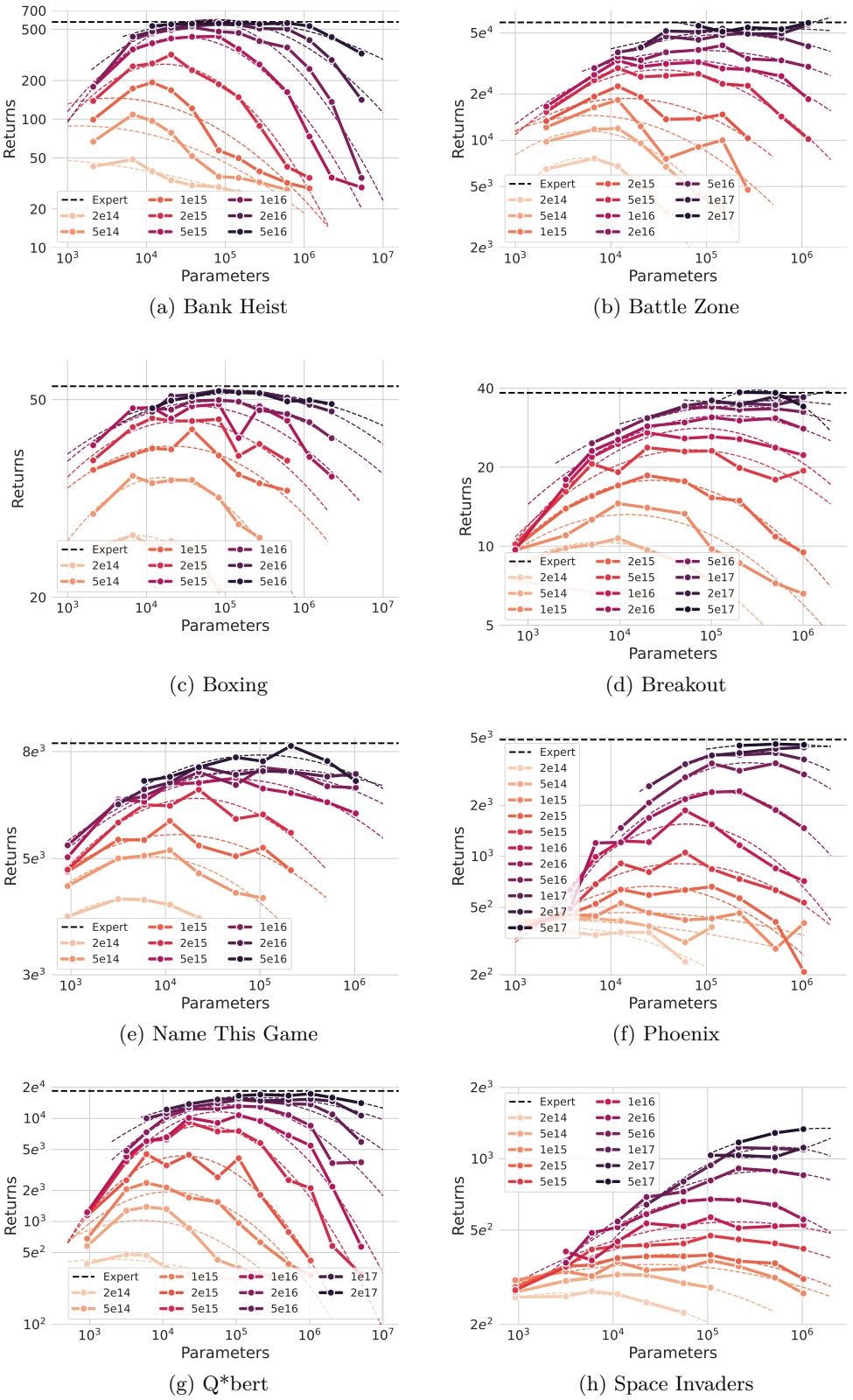

Figure 13: **BC return isoFLOP curves.** Full results for all Atari games.

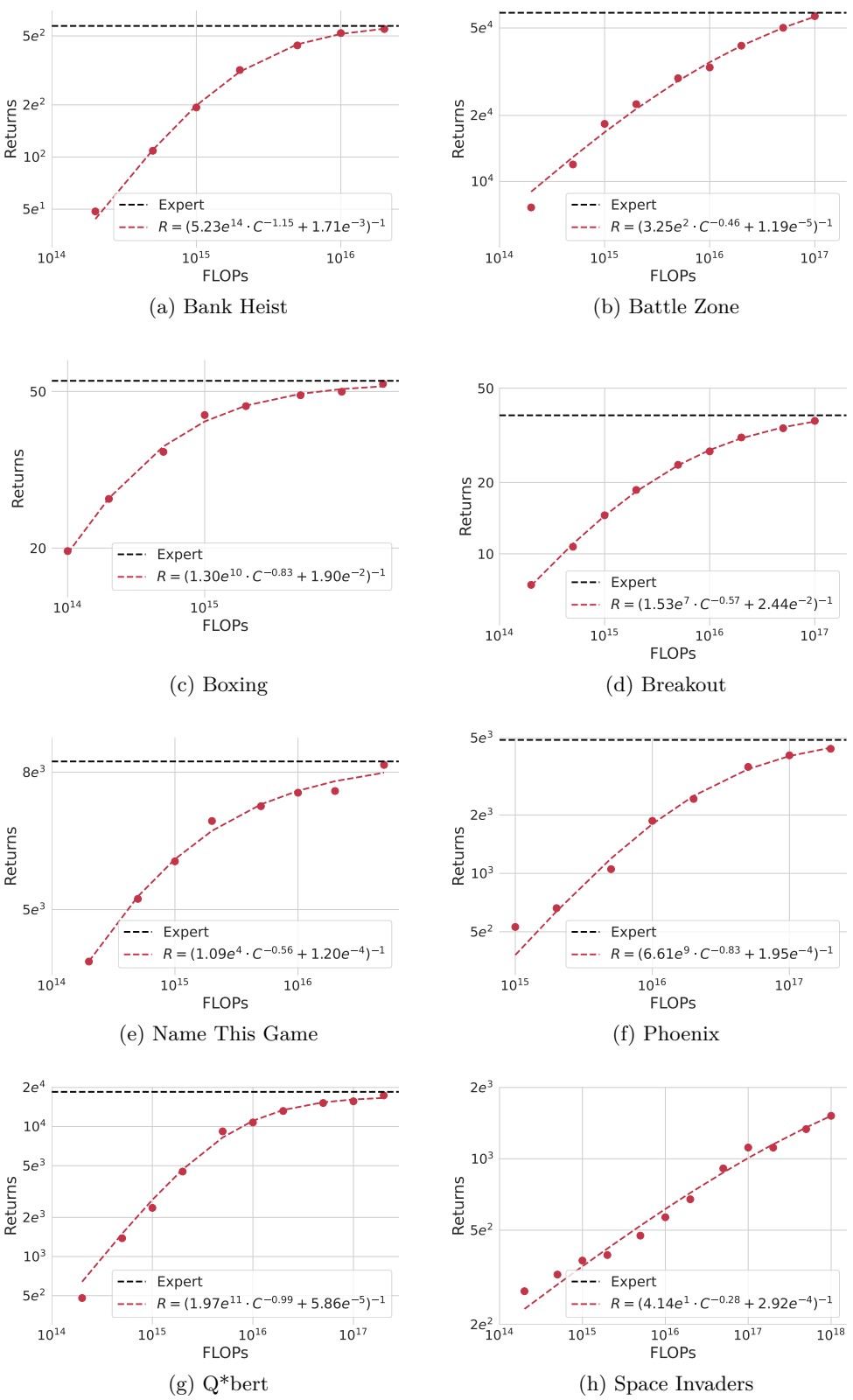

Figure 14: **BC optimal returns vs. FLOPs curves.** Full results for all Atari games.

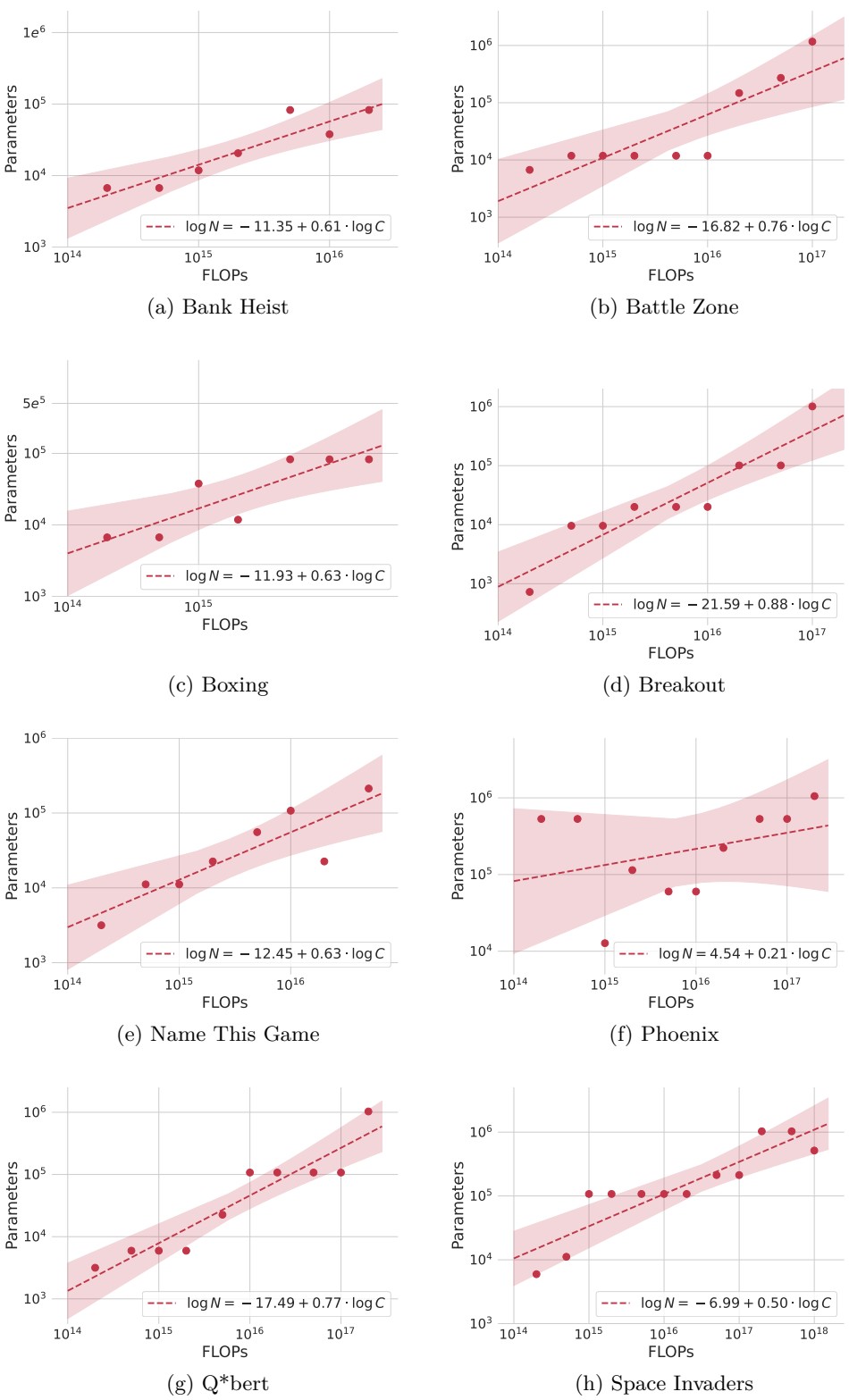

Figure 15: **BC return-optimal parameters vs. FLOPs curves.** Full results for all Atari games. Except For Phoenix, all slope coefficients are statistically significant.

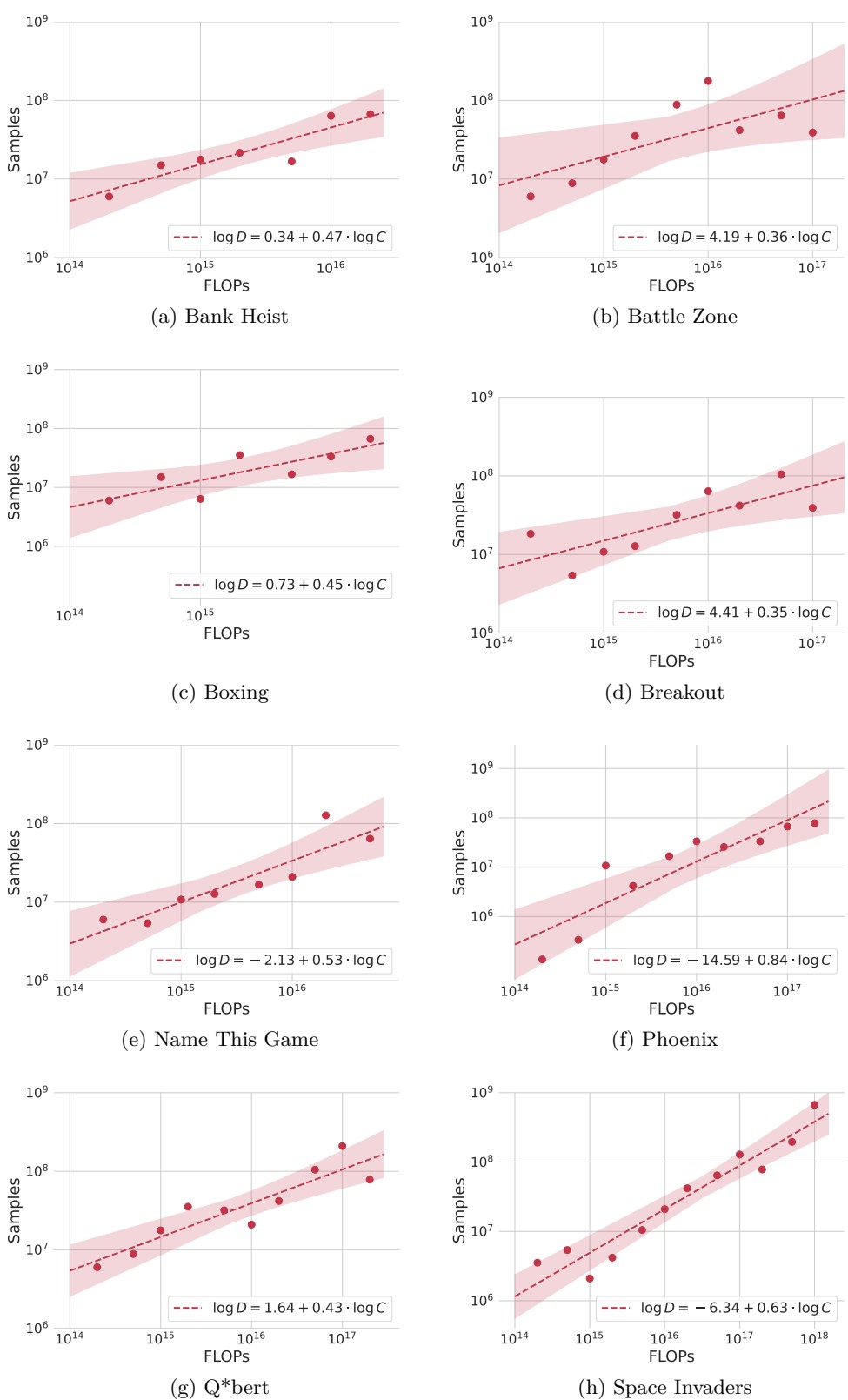

Figure 16: **BC return-optimal samples vs. FLOPs curves.** Full results for all Atari games.

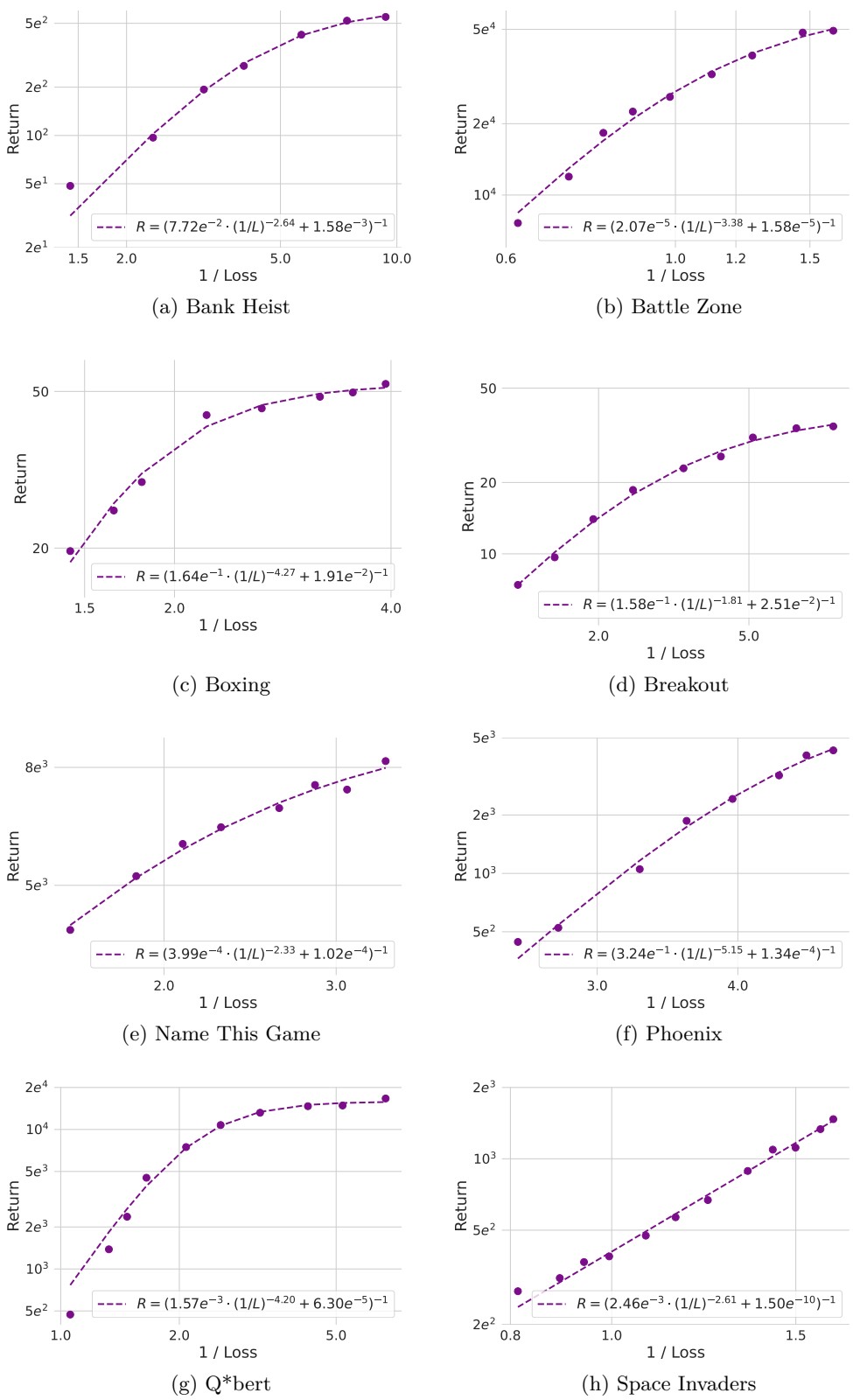

Figure 17: **BC return vs. optimal loss.** Full results for all Atari games.

## K    Descriptions of observation and action space

For Atari, the observation space consists of an 84 x 84 image of pixel values between 0 and 255. The action space is discrete and varies per game. See below for the size of the action space per game:

- Battle Zone: 18
- Q*bert: 6
- Bank Heist: 18
- Boxing: 18
- Breakout: 4
- Name This Game: 6
- Phoenix: 8
- Space Invaders: 6

For NetHack, the observation space consists of 24 x 80 ASCII character and color grids (one for each), along with the cursor position (consisting of 2 coordinates). The action space is discrete and consists of 121 actions for the full game (i.e. `NetHackChallenge-v0`) and 23 actions for the simplified version (i.e. `NetHackScore-v0`).

## L    Reward densities per game

Below we provide for every environment the average number of steps the expert takes before a reward is received from the environment:

1. NetHack: 95
2. Phoenix: 76
3. BattleZone: 20
4. Breakout: 10
5. BankHeist: 8
6. NameThisGame: 5
7. SpaceInvaders: 5
8. Boxing: 5
9. Qbert: 3

