# OpenReview forum: "Scaling Laws for Imitation Learning in Single-Agent Games"
_TMLR — Accepted by TMLR_

### Review · Reviewer_nPQe · 2024-08-29

**Summary Of Contributions:**

The authors conduct an ambitious analysis of imitation learning scaling laws in solved (Atari) and unsolved (NetHack) single-agent games. Their contributions are twofold. First, the authors show that imitation learning metrics such as BC validation loss and return may scale with power-laws as a function of model capacity and training flops in certain environments and certain learning regimes, including several Atari games and the “early-game” portion of NetHack. These scaling relationships echo those previously observed in single-agent RL and in the supervised training of language models. Second, their investigations produce a state-of-the-art, offline-trained agent for NetHack.

**Audience:**

Yes

**Broader Impact Concerns:**

I have no concerns on the ethical implications of this work.

**Claims And Evidence:**

Yes

**Requested Changes:**

The first weakness described above can be addressed by an amendment to Section 6. I believe that this change would strengthen the work.

The second weakness can be addressed by softening the language used to describe the contributions of the paper in the introduction and the conclusion of the paper, aligning it closer with the experimental results of the body. This change is critical to securing my recommendation for acceptance. I believe that there is already a sufficient discussion of possible explanations for the plateau-effect in IL and RL agent mean NetHack returns in the results and discussion sections of the paper.

**Strengths And Weaknesses:**

### Strengths

(1) Timely research question. This is the first work to attempt to characterize the scaling laws of imitation learning ala “Scaling Laws for Neural Language Models” by Kaplan et al for interactive settings, where performance metrics like reward reflect the effects of compounding errors.

(2) Experimentation is thorough. The authors perform evaluations across a large set of single-agent game environments.

(3) The authors achieve state-of-the-art performance for offline-trained, learned agents in NetHack through their experiments.

(4) The writing is clear throughout the paper.


----

### Weaknesses

**Is Mean Return Scaling Well-Defined?**
In single-agent games, reward functions typically reflect external heuristics for progress that have been hand-engineered by game designers. The quality of these heuristics may be variable, even when they rewards are “dense.”

As a result, it is not obvious that the “mean return” metric for which the authors compute scaling laws is necessarily well-defined, unlike the “single-step” metrics of IL validation loss or perplexity.

To be more precise, the observed scaling behaviors of return distributions as a function of model capacity and training FLOPs may be confounded by a shared functional property of the tested reward. Hilton et al. [1] call reward functions “natural performance metrics” if the return of a policy trained with RL (under AlphaZero) “follows a power law in training compute.” In light of this definition, one way to interpret the authors’ return scaling results may be as a confirmation of the IL-variant of this property for the tested Atari and NetHack reward functions. Though the authors cite Hilton et al., they do not discuss this point.

**Some Inconsistency Between Strength of Contribution Claims and Results/Discussion**
In the “Discussion” paragraph of Section 5, the authors find mean return scaling to plateau in NetHack.

This result directly conflicts with the following statement in the “contribution” summary provided in the introduction: “This suggests carefully scaling up model and data size can provide a promising path towards increasingly capable game agents [in NetHack], as well as potentially boost performance in other imitation learning settings.” Similar language can be found in the conclusion.

The authors have convincingly shown through their experiments that IL agents’ NetHack score follow a power-law in training compute + model capacity in the earliest stages of the game. However, their results suggest the opposite overall conclusion of the stated one; namely, that IL may not be a tenable method for beating, or even matching, the performance of the AutoAscend bot. Similar findings have been reported in existing works studying IL + RL for NetHack [3, 4].


---

[1] Hilton, Jacob, Jie Tang, and John Schulman. "Scaling laws for single-agent reinforcement learning." arXiv preprint arXiv:2301.13442 (2023).

[2] Kaplan, Jared, et al. "Scaling laws for neural language models." arXiv preprint arXiv:2001.08361 (2020).

[3] Wołczyk, Maciej, et al. "Fine-tuning Reinforcement Learning Models is Secretly a Forgetting Mitigation Problem." arXiv preprint arXiv:2402.02868 (2024).]

[4] Piterbarg, Ulyana, Lerrel Pinto, and Rob Fergus. "Nethack is hard to hack." Advances in Neural Information Processing Systems 36 (2024).

---

> ### Author Response · Authors · 2024-09-06
> **Rebuttal Part #1**
>
> We thank the reviewer for their thorough and valuable feedback.
>
> > In single-agent games, reward functions typically reflect external heuristics for progress that have been hand-engineered by game designers. The quality of these heuristics may be variable, even when they rewards are “dense.” As a result, it is not obvious that the “mean return” metric for which the authors compute scaling laws is necessarily well-defined, unlike the “single-step” metrics of IL validation loss or perplexity.
>
> First, regarding reward functions reflecting external heuristics progress, we’ve added a footnote about this for NetHack in subsection 4.2 (see text in blue).
>
> Second, mathematically, the mean return is defined as $\mathbb{E} \left[ \sum\limits_{t = 0}^H R (S_t, A_t) \right]$ where the expectation is over the trained imitation learning policy and the MDP, and $H$ is the horizon length. Hence, as long as an environment E provides a reward function R which is defined for all feasible (state, action) pairs, then the mean return is well-defined. This is the case for all the environments we consider in the paper.
>
> > To be more precise, the observed scaling behaviors of return distributions as a function of model capacity and training FLOPs may be confounded by a shared functional property of the tested reward. Hilton et al. [1] call reward functions “natural performance metrics” if the return of a policy trained with RL (under AlphaZero) “follows a power law in training compute.” In light of this definition, one way to interpret the authors’ return scaling results may be as a confirmation of the IL-variant of this property for the tested Atari and NetHack reward functions. Though the authors cite Hilton et al., they do not discuss this point.
>
> This is a great point! While the original version of our paper does already mention natural performance metrics in Section 6 under “Natural performance metrics”, we didn’t explicitly state the reviewer’s suggested interpretation of our work as a confirmation of these natural performance metrics for NetHack and Atari in IL. We have now added this interpretation more explicitly to this same paragraph (see text in red). Let us know if this resolves this point or if the reviewer meant something different.
>
> > In the “Discussion” paragraph of Section 5, the authors find mean return scaling to plateau in NetHack. This result directly conflicts with the following statement in the “contribution” summary provided in the introduction: “This suggests carefully scaling up model and data size can provide a promising path towards increasingly capable game agents [in NetHack], as well as potentially boost performance in other imitation learning settings.” Similar language can be found in the conclusion.
>
> We have softened the language in the intro, abstract, and conclusion. Please refer to the text in red in our revised pdf.
>
> ***Intro:***
> “This suggests carefully scaling up model and data size can provide a promising path towards increasingly capable game agents, as well as potentially boost performance in other imitation learning settings.”
>
> was changed to:
>
> “This suggests carefully scaling up model and data size can provide big boosts to imitation learning performance in single-agent games, helping to narrow the gap between the learner and the expert.”
>
> ***Abstract:***
> “Our work both demonstrates the scaling behavior of imitation learning in a variety of single-agent games, as well as the viability of scaling up current approaches for increasingly capable agents in NetHack, a game that remains elusively hard for current AI systems.”
>
> was changed to:
>
> “Our work both demonstrates the scaling behavior of imitation learning in a variety of single-agent games, as well as helps narrow the gap between the learner and the expert in NetHack, a game that remains elusively hard for current AI systems.”
>
> ***Conclusion:***
> “Our results demonstrate that scaling up model and data size is a promising path towards training increasingly capable agents for single-agent games.”
>
> was changed to:
>
> “Our results demonstrate that scaling up model and data size can provide big boosts to imitation learning performance in single-agent games.”

---

> ### Author Response · Authors · 2024-09-06
> **Rebuttal Part #2**
>
> > However, their results suggest the opposite overall conclusion of the stated one; namely, that IL may not be a tenable method for beating, or even matching, the performance of the AutoAscend bot. Similar findings have been reported in existing works studying IL + RL for NetHack [3, 4].
>
> While we agree with the reviewer our results show that performance peters out before reaching the expert return in NetHack, we nevertheless also show that IL performance can be much better than previously thought or reported [10,11]. In addition, we should never expect IL performance to go beyond the expert performance, since the minimizer of the cross-entropy loss will be a policy that induces the exact same state-action distribution as the expert, and hence accumulates the same statistics (including the same mean return). That being said, the plateau of the scaling laws also varies with the quality of the expert. Hence, while beating or matching AutoAscend might not be possible when AutoAscend is the expert, it *might* be possible when we change the expert to be a collection of humans (using for example the NLD-NAO dataset [10]), who can get scores that are several orders of magnitude higher than AutoAscend.
>
> We hope our rebuttal clarifies the reviewer's confusions and relieves some of their concerns. Please don’t hesitate to follow-up with further questions or discussion.

---

### Review · Reviewer_Fv4n · 2024-08-29

**Summary Of Contributions:**

The paper explores the impact of scaling up model and data sizes on the performance of agents trained using Imitation Learning (IL) in single-agent games. The authors are motivated by the success of scaling in Natural Language Processing (NLP) and aim to investigate whether similar improvements can be achieved in IL, particularly in constrained environments like single-agent games. They conduct experiments on Atari games and the complex game NetHack, discovering that both IL loss and mean return exhibit scaling laws with respect to the compute budget (measured in FLOPs). The study concludes that scaling up models and data size is a viable approach for developing more capable agents, especially in challenging environments like NetHack.

**Audience:**

Yes

**Claims And Evidence:**

Yes

**Requested Changes:**

see weaknesses above.

**Strengths And Weaknesses:**

### Strengths:
1. **Innovative Approach:** The paper takes an innovative approach by applying scaling principles from NLP to the domain of Imitation Learning. This cross-disciplinary inspiration is valuable and opens new avenues for research in IL.
2. **Comprehensive Experiments:** The authors conduct extensive experiments across Atari and NetHack. The variety of environments provides a robust evaluation of their hypothesis.
3. **Clear Findings:** The paper presents clear and actionable findings, demonstrating that IL loss and mean return follow scaling laws with respect to FLOPs. This finding is significant as it provides a new understanding of how compute resources impact IL performance.
4. **State-of-the-Art Results:** In the challenging environment of NetHack, the paper shows that their approach outperforms the previous state-of-the-art (SOTA) by 1.7x, indicating the practical effectiveness of scaling in IL.
5. **Forecasting Compute Requirements:** By deriving scaling laws, the authors not only analyze current performance but also forecast the compute requirements for training optimal IL agents. This predictive capability is useful for future research and practical applications.

### Weaknesses:
1. **Limited Scope of Environments:** The paper focuses on environments with relatively dense rewards. While this is acknowledged as a limitation, it restricts the generalizability of the findings to other environments where rewards might be sparse or noisy.
2. **Dependence on Expert Quality:** The experiments rely on expert agents trained with PPO in environments like Atari and NetHack. The quality of these experts significantly impacts the performance of the IL agents, but the paper does not deeply investigate how variations in expert quality affect the scaling laws.
3. **Scalability Concerns:** While scaling up model and data sizes is shown to be effective, it raises concerns about the practicality of this approach in real-world applications where compute resources are limited. The paper could benefit from a discussion on the trade-offs between scaling and resource efficiency.
4. **Overemphasis on Correlation:** The strong correlation between IL loss and mean return is emphasized, but the paper does not fully explore the causality or potential confounding factors that could influence this relationship. A deeper analysis of these correlations would strengthen the conclusions.
5. **Sparse Discussion on Generalization:** The paper mainly focuses on the immediate performance improvements in specific games. However, it does not provide sufficient discussion on how these findings might generalize to other types of games or tasks within IL. Expanding on this would enhance the impact of the research.
6. **Limited RL Exploration:** The paper briefly extends its exploration to the RL setting by examining the impact of model size and environment interactions on RL performance; however, the discussion remains quite limited. The authors note that compute-optimal agents in RL also benefit from scaling up model size and interactions, but this observation is based on a single RL algorithm (IMPALA) applied to only one game environment. Considering the diverse nature of RL problems with varying dimensionalities and complexities, the paper would benefit from a deeper investigation into how these scaling laws hold across different RL settings, including both online and offline learning scenarios. Additionally, the need for extensive tuning specific to different problem settings is an important consideration that is not fully addressed.

---

> ### Author Response · Authors · 2024-09-06
> **Rebuttal Part #1**
>
> We thank the reviewer for their thorough and valuable feedback.
>
> > Limited Scope of Environments: The paper focuses on environments with relatively dense rewards. While this is acknowledged as a limitation, it restricts the generalizability of the findings to other environments where rewards might be sparse or noisy.
>
> While this is indeed listed as a limitation in Section 6, we note the following:
> - While we do focus on environments with somewhat dense rewards, we note that the reward density still varies quite a bit across our environments, suggesting that our analysis is at least somewhat robust to the density of the reward. Specifically, our most dense environment (Qbert) provides a reward every 3 steps on average, while our most sparse environment (NetHack) only provides a reward every 95 steps on average. **We have added the average steps per reward received for every environment in Appendix K** (see orange text).
> - The restriction to environments with somewhat dense rewards doesn’t mean our settings aren’t useful! **There are plenty of interesting problems in RL which match these settings (i.e. have dense rewards).** Real-world examples of these settings include inventory management [7] and data center cooling [8]. In both problems, we can expect the reward to be pretty dense (see equation 2 in Madeka et al. [7] and equation 3 in Lazic et al. [8]).
>
> > Dependence on Expert Quality: The experiments rely on expert agents trained with PPO in environments like Atari and NetHack. The quality of these experts significantly impacts the performance of the IL agents, but the paper does not deeply investigate how variations in expert quality affect the scaling laws.
>
> ***Note***: The answer below assumes the following definition of “quality of the expert”: the average mean return of the expert in the environment, as compared to random, human, or superhuman baselines. If this is not what the reviewer is referring to by “quality of the expert”, please let us know so we can revise our answer.
>
> First, the results in our paper are already using experts of different qualities for different domains. For example, while AutoAscend (the expert we use for NetHack) is pretty poor compared to a human expert, the expert we use for Breakout is superhuman [9]. Nevertheless, we observe scaling laws in all cases, suggesting that the quality of the expert does not affect the existence of our scaling law. Second, what we do find to vary based on the quality of the expert is the upper bound or ceiling that scaling laws converge to. Specifically, for all Atari games, we find that the scaling laws converge exactly to the mean return of the expert. Hence, we expect that as the expert quality improves or deteriorates for a particular environment, the corresponding scaling law will change to reflect this shift in the ceiling (i.e. it will now plateau at a different mean return).
>
> **In the revised pdf, we added a new discussion paragraph in section 9 about the effect of the expert quality on the scaling laws** (see text in orange).
>
> > Scalability Concerns: While scaling up model and data sizes is shown to be effective, it raises concerns about the practicality of this approach in real-world applications where compute resources are limited. The paper could benefit from a discussion on the trade-offs between scaling and resource efficiency.
>
> We would like to point out that a large part of our paper addresses exactly this question: how do we scale in a compute-optimal (in terms of FLOPs) way? In the setting where compute resources are limited (say we can only use X amount of FLOPs), our paper gives the compute-optimal data and model size such that performance is maximized under the specified FLOP budget. Therefore, we would argue our paper is especially relevant for compute-constrained settings since it gives a way to optimally allocate resources (i.e. model and data size) when constrained by FLOPs.
>
> In addition, **we have added some discussion on data availability in section 6** (see text in magenta). One interesting direction for future work could be extending our results to the data-constrained setting, similar to [6].
>
> > Overemphasis on Correlation: The strong correlation between IL loss and mean return is emphasized, but the paper does not fully explore the causality or potential confounding factors that could influence this relationship. A deeper analysis of these correlations would strengthen the conclusions.
>
> We emphasize that we do not make any causal statement - our claim is purely that loss and return are highly correlated, as shown in Figure 3, which is a purely empirical finding. What specific additional analysis would the reviewer like to see to strengthen our correlation results?

---

> > ### Author Response · Authors · 2024-09-06
> > **Rebuttal Part #2**
> >
> > > Sparse Discussion on Generalization: The paper mainly focuses on the immediate performance improvements in specific games. However, it does not provide sufficient discussion on how these findings might generalize to other types of games or tasks within IL. Expanding on this would enhance the impact of the research.
> >
> > Here are some of the main takeaways from our paper that a practitioner may want to keep in mind when performing BC on their own domain:
> > - **Takeaway #1: When using BC for a new domain, there could be scaling laws describing its performance.** Since our scaling law results empirically hold for a variety of games with diverse characteristics (stochastic, partially observable, pixel-based, etc.), it is possible that behavioral cloning performance on a new environment may exhibit similar (in terms of the functional form, not the exact coefficients) scaling laws. This can in turn allow for making extrapolation predictions about performance benefits when scaling up.
> > - **Takeaway #2: BC’s poor performance relative to the expert might come (in part) due to poor choice of data size and model size.** While there is some prior work arguing for fundamental flaws in BC [3, 4], we propose an alternative perspective: if a practitioner runs BC and it achieves poor performance, this might be due to poor choices with respect to data and model size rather than BC being fundamentally flawed. This is powerful because it means that instead of being led to believe BC simply doesn’t work and spending time on alternative methods, one might be able to stick with BC at the right scale.
> > - **Takeaway #3: One has to be careful about partial observability.** Our results in Section 7 indicate that when performing BC on a new environment, one must always ask the following two questions: (1) does the learner have access to all the features the expert has and (2) does the learner have as much “memory” (i.e. context length) as the expert. If the answer to either of these is “no”, then performance could potentially be improved by addressing these.
> >
> > **We have added a summary of these points to Section 9** under “Beyond single-agent games” (see text in magenta).
> >
> > > Limited RL Exploration: The paper briefly extends its exploration to the RL setting by examining the impact of model size and environment interactions on RL performance; however, the discussion remains quite limited. The authors note that compute-optimal agents in RL also benefit from scaling up model size and interactions, but this observation is based on a single RL algorithm (IMPALA) applied to only one game environment. Considering the diverse nature of RL problems with varying dimensionalities and complexities, the paper would benefit from a deeper investigation into how these scaling laws hold across different RL settings, including both online and offline learning scenarios. Additionally, the need for extensive tuning specific to different problem settings is an important consideration that is not fully addressed.
> >
> > We fully acknowledge the limited discussion for our RL results, but we emphasize that the main topic of our paper is imitation learning, not reinforcement learning. We’ve merely included this section as a way of extending our analysis to connect with the RL community, and as a seed for future work. The fact we get some signal here is particularly exciting since analyzing RL can be more difficult than the supervised setting [5]. It also connects our work to that of Hilton et al. [5] and shows that even in complex environments like NetHack the reward function can act as a natural performance metric. Nevertheless, we agree with the reviewer there’s a lot more to explore in the RL setting, and hope this section can act as a catalyst for future work to do so.
> >
> > We hope our rebuttal clarifies the reviewer's confusions and relieves some of their concerns. Please don’t hesitate to follow-up with further questions or discussion.

---

### Review · Reviewer_j1NP · 2024-08-30

**Summary Of Contributions:**

This work looks at scaling laws for imitation learning in games: exploring a wide range of model sizes and training set sizes, and showing that for a given compute budget the optimal choices are well-fit by an exponential relationship. It looks at behavior cloning of a PPO agent in six Atari games and a rule-based expert system in NetHack, varying the cloned model's size and the data available to the BC training process.

**Audience:**

Yes

**Claims And Evidence:**

Yes

**Requested Changes:**

Repeating the comments from above, I think the paper could be improved by trying to draw a few more conclusions from the data.


Minor comments
What were the selection criteria for the six Atari games?

"imitation learning has powered some of the most impressive feats of AI in recent years. AlphaGo (Silver et al., 2016) used imitation on human Go games"
At least personally, this particular example brings up a slightly awkward association, where the subsequent AlphaZero work noted that IL was a detriment to performance. If the authors have other well known examples, those examples might avoid that association. This is just noting my personal impression: I have no issues with the authors being aware of this possible association and preferring to leave it as is, with the familiarity of AlphaGo as an example being viewed as a larger benefit.

**Strengths And Weaknesses:**

The article is well focused on a specific question. It is mostly experimentally driven, and I think the authors do a reasonable job of exploration: without regard for practicality more data and more domains would always be better, but there are practical limits for any single effort and the experiments used here seem adequate for suggesting a general trend. The best-fit trends proposed by the authors has utility: they show that selecting a choosing a larger set of parameters suggested for a larger compute budget allows the authors to train a better imitation learning agent than seen in prior work.

There were two things that I hoped the article would address, both relating to helping a reader generalize these results beyond NetHack and 6 Atari games.
First, some more discussion of how to interpret the results. The authors do a good job of describing what they're doing, and present the results, but leave analysis and interpretation to the reader. What conclusions should a reader take, and hope to transfer to their own domain? What should a reader take from the similarity in loss scaling between the IsoFLOP data and the parametric fit, but the dissimilarity in return scaling? Should they just use the parametric model to estimate the scaling in their own domain, as the parametric model is slightly more informative? Should a reader expect the demonstrated scaling to transfer at all to a new domain, or is the only general conclusion "bigger is better, with the right parameter / data balance"?

Second, some discussion availability of data. There are a couple of ways that IL in games differs from LLMs. Compared to written text for LLMs, It is much more likely that there is a limited amount of expert data available for a specific game. Another difference is that the expert might be a trained policy: how does cloning the expert scale compared to training the expert?

---

> ### Author Response · Authors · 2024-09-06
> **Rebuttal Part #1**
>
> We thank the reviewer for their thorough and valuable feedback.
>
> > What conclusions should a reader take, and hope to transfer to their own domain? Should a reader expect the demonstrated scaling to transfer at all to a new domain, or is the only general conclusion "bigger is better, with the right parameter / data balance"?
>
> This is a great question! Here are some of the main takeaways from our paper that a practitioner may want to keep in mind when performing BC on their own domain:
> - **Takeaway #1: When using BC for a new domain, there could be scaling laws describing its performance.** Since our scaling law results empirically hold for a variety of games with diverse characteristics (stochastic, partially observable, pixel-based, etc.), it is possible that behavioral cloning performance on a new environment may exhibit similar (in terms of the functional form, not the exact coefficients) scaling laws. This can in turn allow for making extrapolation predictions about performance benefits when scaling up.
> - **Takeaway #2: BC’s poor performance relative to the expert might come (in part) due to poor choice of data size and model size.** While there is some prior work arguing for fundamental flaws in BC [3, 4], we propose an alternative perspective: if a practitioner runs BC and it achieves poor performance, this might be due to poor choices with respect to data and model size rather than BC being fundamentally flawed. This is powerful because it means that instead of being led to believe BC simply doesn’t work and spending time on alternative methods, one might be able to stick with BC at the right scale.
> - **Takeaway #3: One has to be careful about partial observability.** Our results in Section 7 indicate that when performing BC on a new environment, one must always ask the following two questions: (1) does the learner have access to all the features the expert has and (2) does the learner have as much “memory” (i.e. context length) as the expert. If the answer to either of these is “no”, then performance could potentially be improved by addressing these.
>
> **We have added a summary of these points to Section 9 under “Beyond single-agent games”** (see text in magenta).
>
> > What should a reader take from the similarity in loss scaling between the IsoFLOP data and the parametric fit, but the dissimilarity in return scaling? Should they just use the parametric model to estimate the scaling in their own domain, as the parametric model is slightly more informative?
>
> Regarding the dissimilarity in return scaling, we note that scaling law exponents are somewhat hard to measure very accurately and generally come with a substantial amount of uncertainty, as highlighted by past work as well [5]. We can see this too when looking at the confidence intervals for the isoFLOP profiles in Table 1. Therefore, we are hesitant to draw strong conclusions from this dissimilarity, and instead just report it as an empirical finding.
>
> As to how to choose between the two methods, we recommend performing a rolling cross-validation of the relevant power laws for each method, and then picking the one with lowest cross-validation error (e.g. lowest validation RMSE). For simplicity, we did not do this comparison in the paper and instead just ran the forecast based on the isoFLOP method. We have added this explanation to the paper as a footnote on page 8 (see text in magenta).
>
> > Another difference is that the expert might be a trained policy: how does cloning the expert scale compared to training the expert?
>
> While for NetHack, the expert is a rule-based system, for Atari the experts are indeed all trained with PPO. Nevertheless, if we tried to compare these naively we’d be trying to compare scaling laws for BC vs. scaling laws for RL (which have very different objectives). Therefore, we’re not entirely sure how to best go about this. If the reviewer has any suggestions here, we’d love to hear them!
>
> > What were the selection criteria for the six Atari games?
>
> First, we’d like to clarify that we evaluated a total of 8 games, not 6. Please refer to Appendix I for the full set of all 8 Atari results. Second, as specified in subsection 3.1, we chose these games because they have a reward that is at least somewhat dense.

---

> > ### Comment · Reviewer_j1NP · 2024-09-09
> >
> > > Regarding the dissimilarity in return scaling, we note that scaling law exponents are somewhat hard to measure very accurately and generally come with a substantial amount of uncertainty, as highlighted by past work as well [5]. We can see this too when looking at the confidence intervals for the isoFLOP profiles in Table 1. Therefore, we are hesitant to draw strong conclusions from this dissimilarity, and instead just report it as an empirical finding.
> >
> > Table 1 is where the discrepancy seemed to be worth some comment. Comparing the 95% CI is not quite the right statistical test, but it goes from overlapping IsoFLOP / Partametric intervals in the case of loss, to disjoint intervals for return.
> >
> > > While for NetHack, the expert is a rule-based system, for Atari the experts are indeed all trained with PPO. Nevertheless, if we tried to compare these naively we’d be trying to compare scaling laws for BC vs. scaling laws for RL (which have very different objectives). Therefore, we’re not entirely sure how to best go about this. If the reviewer has any suggestions here, we’d love to hear them!
> >
> > There is at least one sample, given the actual expert computation. Where the performance is a horizontal line for the expert, the expert FLOPs might be a vertical line, and the expert FLOPs / parameters could be a single point.
> >
> > > First, we’d like to clarify that we evaluated a total of 8 games, not 6. Please refer to Appendix I for the full set of all 8 Atari results. Second, as specified in subsection 3.1, we chose these games because they have a reward that is at least somewhat dense.
> >
> > Sorry -- yes, there are clearly 8 figures per panel in the appendix, not sure where six came from.
> >
> > What I'm trying to probe at is, how broadly applicable is this? Were these the eight most promising possible games? Were they eight from a set of X candidates, and only 8 were run due to constraints on compute / experimenter time? The authors note that the results might not be as smooth for a game like Montezuma's revenge, so presumably X is at least <= |ALE| - 1. What is X?

---

> ### Author Response · Authors · 2024-09-06
> **Rebuttal Part #2**
>
> > Second, some discussion availability of data. There are a couple of ways that IL in games differs from LLMs. Compared to written text for LLMs, It is much more likely that there is a limited amount of expert data available for a specific game.
>
> We argue this depends a bit on the game, the expert, and the scaling law. If the game has a fast simulator and a computationally cheap expert (as is the case in our paper), then data availability may be less of a problem since we can simply collect more data by rolling out many trajectories in parallel using the expert policy (which is what we did). However, if the game itself is slow or the expert is expensive (e.g. a human), then data availability may indeed become a bottleneck (possibly also depending on how much data has already been collected “for free” on online servers where games are stored, as is the case for human NetHack games). Finally, the specific scaling law for a game dictates how much data we actually need (assuming we want to run in the compute-optimal regime). Hence, if the scaling law indicates we’ll need trillions of data points, we might be data-constrained somewhat irrespective of the computational requirements of running the game and the expert. The opposite is also true, however: for simple games (like Atari), the scaling laws seem to usually indicate we don’t need that much data (< 1B), which means we might still be able to get away with slightly more computationally demanding game simulators and experts. One interesting direction for future work could be extending our results to the data-constrained setting, similar to [6].
>
> **We have added the discussion above to a new paragraph in the limitations section (section 6) called “Availability of data”** (see text in magenta).
>
> > "imitation learning has powered some of the most impressive feats of AI in recent years. AlphaGo (Silver et al., 2016) used imitation on human Go games" At least personally, this particular example brings up a slightly awkward association, where the subsequent AlphaZero work noted that IL was a detriment to performance. If the authors have other well known examples, those examples might avoid that association. This is just noting my personal impression: I have no issues with the authors being aware of this possible association and preferring to leave it as is, with the familiarity of AlphaGo as an example being viewed as a larger benefit.
>
> We thank the reviewer for pointing out this possible association! We will stick with the example for now since we indeed like the familiarity of AlphaGo, but if at some point we think of anything better we’ll make sure to revisit this.
>
> We hope our rebuttal clarifies the reviewer's confusions and relieves some of their concerns. Please don’t hesitate to follow-up with further questions or discussion.

---

> > ### Comment · Reviewer_j1NP · 2024-09-09
> >
> > Regarding the other points here and above, thank you for including the expanded discussion in the manuscript.

---

> > > ### Author Response · Authors · 2024-09-12
> > > **Rebuttal**
> > >
> > > Thank you for the follow-up comments!
> > >
> > > > Table 1 is where the discrepancy seemed to be worth some comment. Comparing the 95% CI is not quite the right statistical test, but it goes from overlapping IsoFLOP / Partametric intervals in the case of loss, to disjoint intervals for return.
> > >
> > > While we don’t have a definitive reason for the cause of this discrepancy (only the uncertainty of scaling exponents hypothesis from earlier), **we’ve updated the pdf with extra discussion on this discrepancy towards the end of sections 4.1 and 4.2** (see text in magenta).
> > >
> > > > There is at least one sample, given the actual expert computation. Where the performance is a horizontal line for the expert, the expert FLOPs might be a vertical line, and the expert FLOPs / parameters could be a single point.
> > >
> > > Thank you for this suggestion! While there is still a discrepancy between counting FLOPs for *cloning* vs. *training* the expert (due to also counting FLOPs for the value network, data collection, etc. when training the expert with RL), we have provided some estimates for the FLOP budgets required to train vs. clone the experts in all Atari games:
> > > 1. **Breakout:** expert training FLOPs = 2.3e16 < ~1e17 to clone the expert
> > > 2. **BattleZone:** expert training FLOPs = 1.6e17 > ~1e17 to clone the expert
> > > 3. **BankHeist:** expert training FLOPs = 2.2e17 > 2e16 to clone the expert
> > > 4. **Boxing:** expert training FLOPs = 1.4e17 > 2e16 to clone the expert
> > > 5. **NameThisGame:** expert training FLOPs = 4.6e17 > 5e16 to clone the expert
> > > 6. **Phoenix:** expert training FLOPs = 2.4e17 > 2e17 to clone the expert
> > > 7. **Qbert:** expert training FLOPs = 1.8e17 < 2e17 to clone the expert
> > > 8. **SpaceInvaders:** expert training FLOPs = 4.6e17 – ?? (we didn’t run high enough FLOP budgets to reach the expert)
> > >
> > > The estimates above suggest that the FLOPs to *clone* the expert tend to be a bit lower than the FLOPs to *train* the expert. Also, while the total number of parameters of all Atari experts is 1.7M, all cloned models reach the expert return with a model size of 1M or lower, echoing distillation results where oftentimes bigger models can be distilled into smaller ones with very similar performance.
> > >
> > > > What I'm trying to probe at is, how broadly applicable is this? Were these the eight most promising possible games? Were they eight from a set of X candidates, and only 8 were run due to constraints on compute / experimenter time? The authors note that the results might not be as smooth for a game like Montezuma's revenge, so presumably X is at least <= |ALE| - 1. What is X?
> > >
> > > The first 4 games were picked from the Atari-5 subset [12], which tries to condense the full set of Atari games to a subset of 5 representative games. We had some trouble training a good (i.e. substantially better than random) expert for Double Dunk (the 5th of the Atari-5 subset), so instead we added 4 other games, chosen at random from pretty much the whole set of Atari games but avoiding any well-known sparse reward games, like Montezuma’s revenge. We didn’t include more than 8 games due to constraints on both compute and experimenter time, but we strongly suspect our results will generalize to most games in the Atari set. Note that the results could even hold true for Montezuma’s revenge as well, but since we didn’t specifically seek out very sparse reward games, we didn’t want to make that claim and instead convey this as a *potential* limitation.
> > >
> > > Let us know if you have any further thoughts or questions!

---

### Review · Reviewer_pNmf · 2024-09-03

**Summary Of Contributions:**

The paper introduces scaling laws for eight Atari games (selected according to the density of rewards) and NetHack in three variants: behavioral cloning (BC) loss, BC returns, and reinforcement learning (RL) returns. Scaling laws are empirical findings that show smooth dependence of the quantity of interest (such as validation loss or rewards), which is typically unknown before training the model on three quantities that are known and controllable at the start of the training (number of model parameters, data size, and computational budget). To arrive at the scaling laws, 12 models were trained for Atari (ranging from 1k to 5M parameters) and 10 for NetHack (ranging from 200k to 200M parameters). The computational budgets span between 1e14 and 5e18 FLOPS. The paper showcases the use of such scaling laws by predicting (via extrapolation) the loss/reward optimal model size, given a fixed number of samples (40B and 55B). The resulting model reaches the score of 7.8K as a Human Monk in NetHack in the offline setting, beating the previous score of `diff History LM` by 1.7$\times$.

**Audience:**

Yes

**Claims And Evidence:**

Yes

**Requested Changes:**

See above.

**Strengths And Weaknesses:**

Strengths:
* The paper is well-written, and an extensive batch of experiments was performed.
* The scaling laws highlight a smooth dependence of loss/rewards on controllable parameters: data size, compute, and the number of parameters.
* Prediction for BC indicates a better-performing NH agent in the offline setting.

Weaknesses:
* The biggest weakness of the paper is the missed opportunity to showcase the application of scaling laws to RL settings (i.e., when RL agents are trained online from scratch). This line of questions is perhaps the most interesting one that could be asked in the paper and which is missing. In particular,
	* Figure 4 is only marginally discussed in the paper.
	* Section 4.3 does not use predictions for reward-optimal models in RL.
	* Appendix H sets a reward target out of reach for the current AI models, i.e., 127k of an average human score (vs 10k of a state-of-the-art AI model). This yields scaling law predictions of model-data size pairs (4.4B, 13.2T) and (67B, 0.93T), respectively. These sizes of models are not only a challenge for current RL methods but also require significant computational resources (in comparison, the biggest model trained for NetHack in the paper has 200M parameters). The Authors use this as an excuse to discard such an analysis as a future work.
	* However, one could ask for optimal-model predictions for RL rewards within the range of the currently available methods (say score in the rage 1k-20k). The answers would either result in new SOTA models or highlight a gap between predictions and the actual results. I would expect there to be a breaking point for the validity of predictions, which would open up an interesting set of research hypotheses. For instance, how good is observed BC/RL behavior in predicting unobserved RL rewards, are there natural barriers e.g., related to computational complexity, what are the conclusion for LLMs, reasoning, etc.
* The paper does not discuss some well-known issues with NetHack's reward structure being misleading.
* Models between training modes and environments are not comparable. Indeed, the NetHack BC version uses transformers, the RL version uses LSTM, and Atari uses CNNs.
* The paper could provide more discussion concerning the shape of scaling-laws curves, i.e., their behavior with respect to the changes in the variables of interest ($N$, $D$, and $C$).

Other questions and remarks:
* Did the authors verify the hypothesis that "for any game score to be a natural performance metric, it needs to be at least somewhat dense". For example, are the results for Atari games that do not satisfy the "density of rewards" criterion not scale smoothly?
* Did the Authors try to explain the difference between BC methods vs Offline RL (as shown in Table 2)? In particular, some guidance in this direction can be gained in [1], which formulates a number of "practical observations."
* Although realized in a different manner, the approach on a high-level resembles upside-down RL [2], where rewards are mapped into actions.
* In Figures 1-2, (b) is not aligned with (c)-(d), which makes it inconvenient to read out the values for extrapolation.

Demonstrating that BC loss, BC rewards, and RL rewards adhere to scaling laws (scale smoothly model size, number of parameters, and compute) is a nice empirical finding. Therefore, both `Yes` in the categories `Claims And Evidence` and `Audience` below have to be treated as borderline positive sentiment. However, the interested audience would be much bigger if the claims had an impact on predicting the agent performance in the RL setting, or the lack thereof, thereby opening a set of interesting questions and hypotheses.

[1] Kumar, A., et al. When Should We Prefer Offline Reinforcement Learning Over Behavioral Cloning?, ICLR 2022.


[2] Schmidhuber, J. Reinforcement Learning Upside Down: Don’t Predict Rewards - Just Map Them to Actions, 2019.

---

> ### Author Response · Authors · 2024-09-06
> **Rebuttal Part #1**
>
> We thank the reviewer for their thorough and valuable feedback.
>
> > The biggest weakness of the paper is the missed opportunity to showcase the application of scaling laws to RL settings (i.e., when RL agents are trained online from scratch). This line of questions is perhaps the most interesting one that could be asked in the paper and which is missing.
>
> While we agree with the reviewer there are plenty of interesting questions around scaling laws for RL, we note that our paper is about scaling laws for *imitation learning*, as made clear in the title as well as throughout the paper. We only have a small subsection about RL as a way of extending our analysis to connect with the RL community, and as a seed for future work. However, this subsection is not the main focus of our work, as is reflected in its length (i.e. it’s only a quarter page).
>
> > The paper does not discuss some well-known issues with NetHack's reward structure being misleading.
>
> We added the following footnote to the paper (see text in blue) when discussing the return scaling laws (subsection 4.2): “While past work has pointed out the NetHack score is not necessarily aligned with winning the game [1], they still recommend using it as a proxy to measure progress in the game.”
>
> > Models between training modes and environments are not comparable. Indeed, the NetHack BC version uses transformers, the RL version uses LSTM, and Atari uses CNNs.
>
> Our paper does not try to make conclusions based on direct comparisons between environments, hence we don’t believe this is a big problem. On the contrary, we see this as a strength since it means our scaling laws are not architecture specific.
>
> > The paper could provide more discussion concerning the shape of scaling-laws curves, i.e., their behavior with respect to the changes in the variables of interest (N, D, and C).
>
> We have provided exact mathematical formulas for our scaling laws in the legends of figures 1, 2, 3, and 4, as well as in equations 2, 3, 5, and 6. Could the reviewer please clarify what additional discussion they would like to see? Maybe some intuitive interpretation of these mathematical forms? If so, we’d be happy to add this to the paper!
>
> > Did the authors verify the hypothesis that "for any game score to be a natural performance metric, it needs to be at least somewhat dense". For example, are the results for Atari games that do not satisfy the "density of rewards" criterion not scale smoothly?
>
> This is a great question! So far, all of the Atari games that we have tested have scaled smoothly both in loss as well as in return (see the full results for all 8 games in Appendix I). However, there are Atari games whose reward functions are much sparser than the ones in the games we use, such as Montezuma’s Revenge. We suspect that results may look less smooth if the reward is really sparse (but maybe not!), and hence in our paper we just focused on environments with at least somewhat dense rewards (i.e. there are intermediate rewards in the game and not just at the very end). However, we have not verified this hypothesis, and leave a deeper investigation to future work.
>
> We have also added a new section in Appendix K (see text in orange) that lists reward densities per game as measured by the average number of steps taken per reward received. We note that the reward density still varies quite a bit across our environments, suggesting that our analysis is at least somewhat robust to the density of the rewards.
>
> > Did the Authors try to explain the difference between BC methods vs Offline RL (as shown in Table 2)? In particular, some guidance in this direction can be gained in [1], which formulates a number of "practical observations."
>
> The focus of our paper is not on explaining the difference between BC methods and offline RL, which is a topic that’s indeed much better suited for the paper the reviewer mentions [2]. However, we have clarified in Appendix B (see text colored in blue) which methods in Table 2 and Table 3 are offline RL and which ones are BC.
>
> > Although realized in a different manner, the approach on a high-level resembles upside-down RL [2], where rewards are mapped into actions.
>
> While BC (the approach we use) and upside-down RL are both supervised methods, there are some key differences. One of the main differences is that BC does not assume the agent has access to the reward, while upside-down RL does. We encourage the reader to take another look at section 2 in our paper, which explains the assumptions and mathematical formalism of BC, which may clarify some of these differences.

---

> ### Author Response · Authors · 2024-09-06
> **Rebuttal Part #2**
>
> > In Figures 1-2, (b) is not aligned with (c)-(d), which makes it inconvenient to read out the values for extrapolation.
>
> We’re not entirely sure what the reviewer means here by “not aligned”. Do they mean the lower and upper limits of the x-axis should be the same? If so, we’re happy to make that adjustment. If not, could they please clarify? Either way, if the reviewer is interested in the values for our extrapolation experiments (for model and dataset sizes), we state them explicitly in section 5. Please refer to the end of the paragraph that starts with “To compute the optimal model size …” for the model and dataset sizes used for extrapolation.
>
> > However, the interested audience would be much bigger if the claims had an impact on predicting the agent performance in the RL setting, or the lack thereof, thereby opening a set of interesting questions and hypotheses.
>
> We respectfully disagree with the reviewer here, as we believe the relative importance or interestingness of scaling laws for RL vs. IL is highly subjective. Members of the RL community might indeed be more interested in scaling laws for RL, while members of the IL community might be more interested in our paper. Therefore, it is not clear to us that the interested audience of our paper is necessarily smaller or bigger than that of a paper studying scaling laws for RL.
>
> We hope our rebuttal clarifies the reviewer's confusions and relieves some of their concerns. Please don’t hesitate to follow-up with further questions or discussion.

---

> > ### Comment · Reviewer_pNmf · 2024-09-10
> >
> > Thank you for your reply. I maintain some of my concerns.
> >
> > * Regarding RL: the paper provides scaling laws for RL (Figure 4) and benchmarks against RL algorithms (Table 2). The predictions from Figure 4 could be used but are not, and the results from Table 2 could be discussed but, again, are not (Table 2 itself is not even referred to in the text).
> >
> > * The paper aims to find a return-optimal model, which is essentially searching for a policy that achieves high rewards. Conceptually, that resembles, e.g., upside-down RL.
> >
> > * Regarding the shape of the scaling laws curves, some examples for NetHarck Figure 1a could be: (a) increasing the budget decreases the loss, (b) increasing the budget and the number of parameters decreases the loss, (c) for a fixed budget, the behavior wrt the number of parameters is parabola-like-shaped, with the loss decreasing up to a certain point and then increasing.
> >
> > * The x-axis of Figure 1-2 (b) and (c)-(d) is not aligned in the sense that the grid of the x-axes differ (e.g., compare the position of 1e18).
> >
> > * Since the combined IL and RL community is larger than the IL community itself, the claim "interested audience would be much bigger if the claims had an impact on predicting the agent performance" does not seem controversial.

---

> > > ### Author Response · Authors · 2024-09-11
> > > **Rebuttal**
> > >
> > > Thank you for the follow-up comments!
> > >
> > > > The predictions from Figure 4 could be used but are not,
> > >
> > > As we have mentioned in the previous rebuttal, the RL analysis is a relatively small part of our paper’s contribution - our main focus is on imitation learning. Therefore, we do not explore forecasting in the RL setting, which probably deserves its own whole paper, though we do investigate it in the IL setting in section 5. However, we would love for future work to take a deep dive in this direction. We hope our initial findings can incentivize researchers in the IL and RL communities to do so.
> > >
> > > > and the results from Table 2 could be discussed but, again, are not (Table 2 itself is not even referred to in the text).
> > >
> > > We agree there could be more discussion about Table 2, thanks for pointing this out! **We’ve updated the pdf to include a discussion about Table 2 towards the end of Section 5** (see text in blue).
> > >
> > > > The paper aims to find a return-optimal model, which is essentially searching for a policy that achieves high rewards. Conceptually, that resembles, e.g., upside-down RL.
> > >
> > > While we’re happy to chat more with the reviewer about a potential connection they see with upside-down RL, we first would like to check whether there is a specific concern here that the reviewer would like us to fix in the paper? Do they feel we should mention a possible connection with upside-down RL? Or maybe something else?
> > >
> > > > Regarding the shape of the scaling laws curves, some examples for NetHarck Figure 1a could be: (a) increasing the budget decreases the loss, (b) increasing the budget and the number of parameters decreases the loss, (c) for a fixed budget, the behavior wrt the number of parameters is parabola-like-shaped, with the loss decreasing up to a certain point and then increasing.
> > >
> > > We agree with the reviewer that intuitive descriptions like these would be helpful! Therefore, **we’ve updated the pdf to add such descriptions in sections 4.1 and 4.2** (see text in blue).
> > >
> > > > The x-axis of Figure 1-2 (b) and (c)-(d) is not aligned in the sense that the grid of the x-axes differ (e.g., compare the position of 1e18).
> > >
> > > Thanks for pointing this out! **We’ve fixed this and updated the pdf.**
> > >
> > > > Since the combined IL and RL community is larger than the IL community itself, the claim "interested audience would be much bigger if the claims had an impact on predicting the agent performance" does not seem controversial.
> > >
> > > First, our claims do have an impact on predicting the agent performance (in IL), as we show in Section 5. Second, we had previously understood the reviewer to argue that a paper about scaling laws in RL would have a much bigger audience than one about scaling laws in IL. It is this argument that we disagree with, since we believe the relative importance or interestingness of scaling laws for RL vs. IL is highly subjective. However, from the new response above, we now understand the reviewer is arguing that a paper about scaling laws for IL + RL has a bigger audience than a paper just about scaling laws for IL. We agree this is not a controversial claim, however we also argue this is a somewhat trivial criticism, since this argument could be made for any paper writing about any topic Y - a paper that included topic Y + another topic X would always have a broader audience.
> > >
> > > Let us know if you have any further thoughts or questions!

---

### Decision · Action_Editor_h8TT · 2024-11-19

**Recommendation:** Accept with minor revision

**Comment:**

Motivated by the work in the language modeling domain, where it is shown that there are scaling laws that govern how the number of compute FLOPS should scale with the dataset size, this paper explores the existence of scaling laws in imitation learning (behavioral cloning). Several cases with the scaling behaviors are shown for both Atari games and NetHack environment and the results are briefly extended to reinforcement learning. Overall, the paper presents rich experiments that could be useful for future investigation. However, the reviewers and myself find that the results still don't capture sufficient insights that could have potentially been extracted from these experiments. Unlike language, IL is very much dependent on the environment, and therefore the scaling laws in this paper are unlikely to generalize beyond the environments that are studied in this paper in their current form. I find this is not a blocker for publication at TMLR but I think the authors ought to release all the generated data so that future work can look into extracting more insights from these experiments so that hopefully insights that are generalizable for solving imitation learning problems could be extracted from them. The paper is recommended to be accepted with minor revisions that respond to these comments and those of the reviewers as much as possible, quoted below. Congratulations to the authors!

**Reviewer pNmf**: It feels that with the results of the paper, the Authors could offer the readers more insights. The reluctance of using RL scaling laws for prediction, indicates that there might be a fundamental gap between IL and RL setups, an observation that could lead to multiple interesting research directions.

 **Reviewer Fv4n**: To help the paper reach its full potential, I suggest the authors address the following in future work:

* Conduct a more detailed analysis of the IL loss and mean return correlation, and explore how the results generalize across different domains.
* Clarify the comparability of models across different environments and training architectures.
* Expand the application of scaling laws to RL settings to explore their broader relevance.

**Reviewer nPQe**: While the results in the work are novel, in that no prior work has not examined scaling laws for imitation learning settings, the tested environments cover only the extremes of the task space of interest -- NetHack is extremely difficult, while Atari is relatively simple.

**Reviewer j1NP**: Are the claims and made in this submission supported? Yes, because the paper is very hesitant to make any claims. Like other reviewers I want to see something more. Experiment run, analysis done to the point of data fitting and standard error, but hesitance to draw any conclusions. I think the experiments are well done, but I still remain unsure what I or other readers should do with the results. Even directly asking what should readers take away from this paper, the authors' clarified takeaway message is (1) performance might scale with data or model size, maybe (2) maybe your behavior cloning run would a different amount of data or a different model size (3) missing feature hurts agent performance.

**Reviewer j1NP**: It would be nice to get a more precise statement than "like Montezuma's Revenge" but the selection process described in the author response does not make me think there's something hiding in the selection of games. If I was more inclined to recommend the article be published in close-to-current state, I guess I might have been more inclined to get rid of that last ambiguity and ensure that same description of game selection was in the article text.

**Audience:**

The paper is definitely of broad interest to the community.

**Claims And Evidence:**

While the claims of the paper are supported by the evidence provided, the reviewers and I think that the claims are slightly on the weaker side.

---

> ### Author Response · Authors · 2024-12-19
> **Camera Ready Version Uploaded**
>
> Dear Action Editor,
>
> We're very happy to hear our paper was accepted, and we'd like to let you know we've uploaded the camera ready version which includes some of the requested revisions. Specifically, we have made the following efforts:
>
> 1. From the action editor:
> > but I think the authors ought to release all the generated data so that future work can look into extracting more insights from these experiments so that hopefully insights that are generalizable for solving imitation learning problems could be extracted from them.
>
> We have uploaded the full NetHack dataset to a public dropbox folder so that future work can easily access it. Note that this includes over 3000 zip files with about 18TB worth of NetHack trajectories from the AutoAscend expert bot. Link to the full dataset: https://www.dropbox.com/scl/fo/n53h78ls689rn944cnkz6/AI-21qxU1qptyJMi-zKYNyQ?rlkey=ryjhi2gqjg535xu0q8goczqn1&st=szfonmhh&dl=0
>
> In addition, we have made available all code and experimental data that will allow others to further build on our work. This includes all raw data that was used to generate all the graphs in the paper. Link to the code: https://github.com/princeton-nlp/il-scaling-in-games
>
> 2. From reviewer Fv4n:
> > Clarify the comparability of models across different environments and training architectures.
>
> We have included a small new section in the limitations section of our paper that further clarifies this point.
>
> 3. From reviewer j1NP:
> > It would be nice to get a more precise statement than "like Montezuma's Revenge" but the selection process described in the author response does not make me think there's something hiding in the selection of games. If I was more inclined to recommend the article be published in close-to-current state, I guess I might have been more inclined to get rid of that last ambiguity and ensure that same description of game selection was in the article text.
>
> We have added a new section in the appendix (Appendix F Game Selection for Atari Games) that lays out our game selection process for Atari in detail. We have also made sure to add a reference to this new appendix in the main text.
>
> Regarding the remaining revision comments, we found it hard to extract further actionable insights that also do not require substantial new experimental efforts. However, if the action editor has further suggestions here, we would be happy to try and incorporate them.
>
> Thank you.

---

> > ### Comment · Action_Editor_h8TT · 2024-12-19
> > **Approved**
> >
> > Dear authors,
> >
> > I think the revisions are satisfactory. While this paper may not provide all the actionable insights for further understanding imitation learning problems, I think it takes a nice step forward.
> >
> > Congratulations again!\
> > Ahmad